

# An Accurate and Homogeneous Altimeter Sea Level Record from the ESA Climate Change Initiative

Jean-François Legeais[1], Michaël Ablain[1], Lionel Zawadzki[1], Hao Zuo[2], Johnny A. Johannessen[3], Martin G. Scharffenberg[4], Luciana Fenoglio-Marc[5], M. Joana Fernandes[6,7], Ole Baltazar Andersen[8], Sergei Rudenko[9,10], Paolo Cipollini[11], Graham D. Quartly[12], Marcello Passaro[9], Anny Cazenave[13], Jérôme Benveniste[14]

[1]Collecte Localisation Satellite (CLS), 31520 Ramonville-Saint-Agne, France
[2]European Centre for Medium-Range Weather Forecasts, Reading, UK
[3]Nansen Environmental and Remote Sensing Center (NERSC), Bergen, Norway
[4]University of Hamburg, Hamburg, Germany
[5]University of Bonn, Bonn, Germany
[6]Faculdade de Ciências, Universidade do Porto, 4169-007, Porto, Portugal
[7]Centro Interdisciplinar de Investigação Marinha e Ambiental (CIIMAR), 4450-208 Matosinhos, Portugal
[8]DTU Space, 2800 Kongens Lyngby, Denmark
[9]Deutsches Geodätisches Forschungsinstitut, Technische Universität München, 80333 Munich, Germany
[10]Helmholtz Centre Potsdam - GFZ German Research Centre for Geosciences, 14473 Potsdam, Germany
[11]National Oceanography Centre, Southampton, SO14 3ZH, UK
[12]Plymouth Marine Laboratory, Plymouth, PL1 3DH, UK
[13]LEGOS, 31400 Toulouse, France
[14]ESA/ESRIN, 00044 Frascati, Italy

*Correspondence to*: Jean-François Legeais (jlegeais@cls.fr)

**Abstract.**

Sea Level is a very sensitive index of climate change since it integrates the impacts of ocean warming and ice mass loss from glaciers and the ice sheets. Sea Level has been listed as an Essential Climate Variable (ECV) by the Global Climate Observing System (GCOS). During the past 25 years, the sea level ECV has been measured from space by different altimetry missions that have provided global and regional observations of sea level variations. As part of the Climate Change Initiative (CCI) program of the European Space Agency (ESA) (established in 2010), the Sea Level project (SL_cci) aimed at providing an accurate and homogeneous long-term satellite-based sea level record. At the end of the first phase of the project (2010-2013), an initial version (v1.1) of the sea level ECV has been made available to users (Ablain et al., 2015). During the second phase (2014-2017), improved altimeter standards have been selected to produce new sea level products (called SL_cci v2.0) based on 9 altimeter missions for the period 1993-2015 (https://doi.org/10.5270/esa-sea_level_cci-1993_2015-v_2.0-201612). Corresponding orbit solutions, geophysical corrections and altimeter standards used in this v2.0 dataset are described in details in Quartly et al. (2017). The present paper focuses on the description of the SL_cci v2.0 ECV and associated uncertainty and discusses how it has been validated. Various approaches have been used for the quality assessment such as internal validation, comparisons with sea level records from other groups and with in-situ measurements,





sea level budget closure analyses and comparisons with model outputs. Compared to the previous version of the sea level ECV, we show that use of improved geophysical corrections, careful bias reduction between missions and inclusion of new altimeter missions lead to improved sea level products with reduced uncertainties at different spatial and temporal scales. However, there is still room for improvement since the uncertainties remain larger than the GCOS requirements.

Perspectives for subsequent evolutions are also discussed.

## 1 Introduction

Present-day Global Mean Sea Level (MSL) rise primarily reflects the amount of heat added in the ocean and land ice melt in response to anthropogenic global warming (e.g. IPCC, 2013; von Schuckmann et al., 2016). Accurate monitoring of the sea level is required to better understand its variability and distinguish between natural and anthropogenic forcing factors at the

origin of observed changes. It also allows validation of the climate models developed for projecting future changes, as the models are supposed to correctly reproduce present-day and recent-past changes. Since 1993, satellite altimetry missions deliver accurate sea level measurements, allowing the monitoring of sea level variations at different spatial and temporal scales (Pujol et al., 2016; Ablain et al., 2017; Escudier et al., 2017). About a decade ago, the Global Climate Observing System (GCOS) defined a list of key parameters of the Earth system, or 'Essential Climate Variables' (ECVs) that need to be

accurately monitored in order to meet the needs of the climate change community (Bojinski et al., 2014). To respond to this need for climate-quality satellite data, the European Space Agency (ESA) developed the 'Climate Change Initiative' (CCI) program. This program aims at realizing the full potential of the long-term global Earth Observation archives from satellites as a significant and timely contribution to the ECV databases for climate modelers and researchers. Sea level is one of the listed ECVs of the CCI program. During the first phase (2010-2013) of the sea level CCI project (SL_cci), a first version of

the ECV over the 1993-2010 time span was produced and distributed to the users. Details of the production and validation protocol of this ECV are described in Ablain et al. (2015).

Within the second phase of the project (2014-2017), the objective was not only to extend the length of the sea level record by additional 5 years (2010-2015) but also to provide a full reprocessing of the sea level ECV during the altimetry period thanks to the development and selection of new altimeter algorithms to improve the ECV accuracy, stability and homogeneity. The

details of the orbit solutions, the geophysical corrections, the altimeter standards and processing algorithms selected for the production of this v2.0 ECV are fully described in Quartly et al. (2017).

This paper intends to describe the SL_cci v2.0 ECV and present some validation results obtained through different approaches. After a short description of input data and altimeter standards used in the production system (Sect. 2), a presentation of the v2.0 SL_cci products is provided in Sect. 3. The quality assessment of the ECV is described in Sect. 4.

The consistency with the sea level records provided by other groups has been checked and comparisons were also performed with in-situ tide gauge measurements and combined Argo-based steric and GRACE-based barystatic sea level data. Additional validations based on a sea level budget closure approach and comparisons with the output from high resolution





ocean models are also presented (Sect. 5). The sea level errors and uncertainties are discussed in Sect. 6 before discussing the perspectives of evolution of the sea level products.

## 2 Input data and altimeter standards

The estimation of the altimeter-based sea level is currently based on measurements from many satellite missions (spanning over more than 20 years). The input data used for the production of the first version of the SL_cci ECV (v1.1) were derived from TOPEX/Poseidon, Jason-1, Jason-2, ERS-1, ERS-2, GeoSat Follow-On (GFO) and Envisat satellites. The first 3 missions fly along the so-called "reference orbit", sampling the ocean between 66°S and 66°N. The remaining missions have a higher orbital inclination, providing improved ocean sampling and giving near-complete coverage of the Arctic. A weakness of the v1.1 ECV is the limited number of satellite altimeters used in the production system. In particular, new altimeter missions (e.g., SARAL/Altika, Cryosat-2) have not been included when the temporal extensions of the dataset have been produced. Thus a single satellite (Jason-2) –instead of two to four satellites at the same time during the altimetry era- has contributed to the v1.1 sea level record after the loss of Jason-1 in June 2013. It has affected the ECV in terms of reduced spatial coverage (no measurements north of 66°N) but also in terms of variance due to a deterioration of the sampling of the ocean. These elements have been improved in the SL_cci v2.0 ECV since new altimeter missions (SARAL/AltiKa and CryoSat-2) have been additionally included in the production system, covering the period Jan. 1993 to Dec. 2015 (see Fig. 1 of Quartly et al., 2017).

The ESA CCI objectives put strong emphasis on developing homogeneous datasets with long-term consistency, which not only necessitates stability for the duration of individual altimeter missions, but also great care in minimizing bias between missions. This has significantly impacted the SL_cci project since a particularity of altimetry is that the sea level estimation requires implementation of many different algorithms to provide corrections for orbits, atmospheric delays, tides and sea surface effects (Fu & Cazenave, 2001; Ablain et al., 2017; Escudier et al., 2017). Many different solutions have been adopted over the past decades for each altimeter standard, either developed by the SL_cci consortium or provided by external projects. To select the most appropriate one to ensure homogeneity and stability of the sea level product, the SL_cci project held an Algorithm Selection Meeting in Nov. 2015, during which the latest algorithms were independently evaluated and validated thanks to a formal validation protocol. The associated round robin data packages (RRDP) showing the impact analysis of each standard and the output of the meeting are available on the SL_cci website (http://www.esa-sealevel-cci.org/PublicDocuments), with a synopsis of the comparisons given in Quartly et al. (2017). Many of the applied corrections have been revised, in particular modeled orbits due to time-variable gravity (Rudenko et al., 2014; Couhert et al., 2015, Rudenko et al., 2016), satellite attitude, macromodels and tropospheric correction models for DORIS observations (Rudenko et al., 2017), modifications to the wet tropospheric correction based on combined GNSS and radiometer data sets (Fernandes et al., 2015) and the latest changes in ocean tide (FES2014) and "pole tide" (Desai et al., 2015).



## 3 Description of the SL_cci v2.0 ECV

The SL_cci v2.0 products are based on multi-mission sea level measurements. They are provided as a database of different elements, referenced with the following DOI: 10.5270/esa-sea_level_cci-1993_2015-v_2.0-201612. The different products available for the users are the following:

(1) The Fundamental Climate Data Record (FCDR) is the along-track sea level anomaly (SLA) derived from the 9 altimeter missions, available at 1Hz resolution corresponding to a ground distance of ~7 km. The files include a quality control indicator to remove spurious measurements and all altimeter standards applied in the SLA calculation (geophysical corrections, mean sea surface). In addition, information derived from the inter-mission sea level cross-calibration is provided in order to remove global and regional biases and to homogenize long spatial scale errors (e.g. due to orbit calculation). In

addition to the FCDRs, gridded values of the altimeter Dynamic Atmospheric Correction (DAC) forced by the ERA Interim reanalysis used for the production of the SL_cci products (Carrère et al., 2016) are also available to the users. This may be of interest when comparing altimeter data with in-situ measurements from tide gauges since both dataset should be corrected for the same atmospheric correction.

(2) The SL_cci ECV consists of monthly gridded time series (Jan. 1993 – Dec. 2015) of multi-mission merged SLA at a

spatial resolution of 1/4°. The SL_cci v2.0 ECV has been generated thanks to the CNES/CLS DUACS production system with the same procedures as for the previous version v1.1 (Ablain et al., 2015) (except that the grids have been shifted by half a pixel in v2.0). The main processing steps (developed in Ablain and Legeais, 2014) are: acquire and pre-process data, perform input check and quality control, inter-calibrate and unify the multi-satellite measurements and generate along-track and gridded merged products (based on a monthly optimal interpolation). A land / sea mask derived from the LandCover_cci

project has been applied to all sea level grids. The long-term stability and large scale changes of the SL_cci v2.0 dataset are built upon the records from missions of the reference orbit (TOPEX/Poseidon, Jason-1 and Jason-2 for that period). All have the same 9.92-day orbital cycle at high altitude (1336 km), making satellite trajectories less sensitive to higher order terms of the Earth's gravity field. Data from the other missions (also called complementary missions) contribute to improve the sampling of mesoscale processes, provide the high-latitude coverage and increase the product accuracy. More details on the

SL_cci ECV processing are provided in Quartly et al. (2017) and additional general information on the altimeter data processing can be found in Pujol et al. (2016).

(3) Ocean indicators are derived from the SL_cci ECV: Global Mean Sea Level (GMSL) time series, regional grids of sea level trends (Fig. 1) and maps of the amplitude and phase of the annual (Fig. 2) and semi-annual signals during the period available.

(4) In addition to the SL_cci ECV, the along-track inter calibrated sea level measurements of each mission (level 3 of the altimeter processing) are also available for the users. The included information is the filtered and subsampled valid SLA, where long wavelength biases have been removed to make observed sea level measurements homogeneous and consistent



between the 9 altimeter missions. These data are the input measurements of the mapping procedure and can be used for data assimilation in ocean models for instance.

(5) Improving the quality of the Arctic sea level record has also been one of the key regional foci during the SL_cci project. This has led to two new Arctic sea level records available for the users. One is based on improved waveform classification and retracking, applied on the Envisat and SARAL/AltiKa missions (Poisson et al., 2017). The second is based on the ALES+ altimeter retracking and is derived from the ERS-1 & 2, Envisat and CryoSat-2 missions (Passaro et al., 2017).

## 3 Quality Assessment at Climate Scales

The validation of the reprocessed SL_cci v2.0 ECV has been carried out distinguishing different temporal signals at global and regional scales.

### 3.1 Global MSL long term evolution

The GMSL trend derived from the SL_cci ECV v2.0 during the period 1993-2015 amounts to 3.3 mm/yr +/- 0.5 mm/yr with a confidence interval of 95%. Comparison to in-situ tide gauge measurements (Valladeau et al., 2012) indicates that no drift is found in this reprocessed altimeter dataset given the uncertainty of the method (Prandi et al., 2015). The GMSL trend derived from the reprocessed dataset is the same as the one derived from the ensemble mean of GMSL from other altimeter groups (Fig. 3). When compared to the previous v1.1 ECV, no trend difference is observed during the common reduced period (1993-2014) (Fig. 4). However, at decadal time scale, the v2.0 GMSL trend are significantly different than the v1.1 trend (by -0.2 mm/yr during 1993-2003 and +0.2 mm/yr during 2004-2014, see Fig. 4). This is mainly due to the use of the level 2 GNSS Path Delay Plus (GPD+) wet troposphere correction (Fernandes and Lázaro, 2016) for all missions in the v2.0 (except for GFO) (see Quartly et al., 2017 for more details). This is due to the fact that all radiometers used in GPD+ V2.0 have been calibrated against the Special Sensor Microwave Imager and the Special Sensor Microwave Imager/Sounder, due to their known stability and independent calibration (Wentz, 2013).

For the v2.0 GMSL, the same trend of 3.3 mm/yr is found for the (1993-2003) and (2005-2015) altimetry decades, indicating a steady rise of the GMSL. However, several recent studies using different approaches suggest that an instrumental drift has affected the TOPEX-A altimeter measurements during 1993-1998 (Valladeau et al., 2012; Watson et al., 2015; Dieng et al., 2017 and Beckley et al., 2017). Such a drift (~1.5 mm/yr in terms of GMSL trend) has not been corrected in the SL_cci v2.0 ECV. Applying the suggested correction would lead to a reduced GMSL trend during the total period (3.0 mm/yr) and a greater GMSL rate of rise during the second half of the altimetry era compared to the first half, highlighting that the GMSL rise is accelerating.



### 3.2 Inter annual signals

The mean differences between the SL_cci v2.0 and SL_cci v1.1 are related to the different Mean Sea Surface (MSS) used in both dataset (DTU15 and DTU10 respectively, see Quartly et al., 2017). At interannual time scale, differences arise because of different mean reference periods used to compute the MSS (the period during which sea surface height measurements have been averaged). The reference period of the MSS DTU10 is 1993-2008 (15 years) whereas it is 1993-2012 (20 years) for the MSS DTU15. This is of major importance in the context of data assimilation in ocean models. The users interested in changing the reference period of the dataset can refer to the procedure described in annex A of Pujol et al. (2016). In addition to the reference period of the MSS, it is worth noting that a convention has been applied on the v2.0 sea level grids so that the averaged sea level during year 1993 is set to zero.

In the v1.1 SL_cci ECV, a 1 mm jump was found in the GMSL around mid 2008. This is related to irregularities present in the Jason-1 radiometer data used to compute the GPD+ wet troposphere correction, the one present in the Radar Altimeter Database System (RADS), enhanced near the coast using the methodology described in Brown (2010). The corresponding error was accounted for via a GMSL bias between Jason-1 and Jason-2 and was propagated to Jason-2 over the whole period. This error has been reduced in the v2.0 reprocessing (Fig. 4).

The interannual variations of the SL_cci v2.0 GMSL remain in the envelope of the ensemble mean of the GMSL data from other groups (Fig. 3), which illustrates the homogeneity of the processing of the satellite measurements at these time scales. More validation details at inter annual time scale are provided by comparison with other GMSL products (section 5).

### 3.3 Seasonal cycle

The regional amplitude and phase of the annual cycle of the sea level ECV v2.0 are illustrated on Figure 2 and at global scale, the assessment of the GMSL annual cycle from the v2.0 ECV is based on the monthly climatology during the period of the sea level record (see Figure 5). The resulting signal displays a smoother sinusoidal cycle than the one derived from the ensemble mean of the monthly climatology derived from other products. Given the expected shape of the sea level annual cycle (Chen et al., 1998; Legeais et al., 2016a) and the long length of the record (that filters out the potential peaks during this period), this suggests an improved estimate of the seasonal signal in the reprocessed SL_cci ECV. Compared to the v1.1 ECV, a small difference is observed in terms of the amplitude (1 mm). It is assessed by the comparison with tide gauge measurements (Valladeau et al., 2012). The amplitude of the annual cycle of the sea level difference with in-situ data reaches 2.4 mm with the v1.1 ECV and is reduced to 1.6 mm with the v2.0 ECV, suggesting that the annual signal is better retrieved with the reprocessed dataset. This observed difference is related to the changes in the level 2 altimeter standards involved in the ECV production, the main contributors being the orbit solutions (Couhert et al., 2015; Rudenko et al., 2017) and the GPD+ wet troposphere correction (Fernandes and Lázaro, 2016) used for the different altimeter missions (Quartly et al., 2017). The new pole tide correction (Desai et al., 2015) also affects the amplitude of the annual cycle. The comparison with external independent data confirms that it leads to an improved sea level estimation compared to the v1.1 ECV (Wahr,



1985). This is illustrated by the Taylor diagram (Taylor, 2001) in Fig. 6 which compares the amplitude of the annual cycle of the Envisat and Jason-1 sea level computed with both pole tide corrections, to the sum of the steric dynamic height anomalies derived from the Argo in-situ network and the GRACE ocean mass contribution (grey dot on the x-axis). An increased correlation between both dataset and a reduced variance of the difference is obtained with the new correction.

## 3.4 Regional MSL trends

The regional sea level trends during 1993-2015 (Fig. 1) can considerably deviate from the global mean (values range spatially between -5 and +5 mm/yr around the 3 mm/yr global estimate). Over this 23-year long time span, this is essentially due to non uniform thermal expansion (Stammer et al., 2013), in response to natural internal climate variability (Meyssignac et al., 2012; Palanisamy et al., 2015a, 2015b; Han et al., 2017). However, in some regions, like the southern oceans, anthropogenically forced signal is already probably emerging. The regional sea level trends during 1993-2015 exhibit large-scale variations with regions of almost no sea level change and others with amplitudes reaching up to +8-10 mm/yr such as in the western Tropical Pacific Ocean (Fig. 1). In this area, trends are mainly of thermosteric origin (Legeais et al., 2016b; Meyssignac et al., 2017) in response to increased easterly winds during the last two decades associated with the decreasing Interdecadal Pacific Oscillation (IPO)/Pacific Decadal Oscillation (e.g. Han et al. 2010; McGregor et al. 2012; Merrifield et al. 2012; Palanisamy et al. 2015a; Rietbroek et al. 2016).

The regional trend differences between the SL_cci v2.0 and v1.1 display values ranging between -2.0 mm/yr and +2.0 mm/yr (Fig. 7). The large scale differences are explained by the differences of altimeter standards used in both versions and the orbit solutions are the main contributor (see more details in Quartly et al., 2017). The small scale differences observed over the global ocean are related to the difference in the satellite constellation between both versions of the ECV. Indeed, CryoSat-2 and SARAL/AltiKa missions are used after 2012 in v2.0 and were not included in v1.1. This means that sampling of the ocean is not the same in both datasets: the empty interleaved spaces between Jason-2 tracks in the v1.1 ECV have been sampled with CryoSat-2 and SARAL/AltiKa in the v2.0 ECV, which directly affects the trend differences, especially in regions of high ocean variability.

### 3.5 Mesoscale signals

The SLA variance provides an estimate of the sea level variability referenced to the Mean Sea Surface used for the SLA calculation. The global SLA variance differences between SL_cci v2.0 and v1.1 time series is on average of +3 cm2 over the common period indicating that more variability is observed in the reprocessed ECV (even when removing the seasonal cycle). This change in the SLA variance is explained by several factors: inclusion of new missions (CryoSat-2 and SARAL/AltiKa) in the v2.0 ECV, leading to an improved mesoscale estimation and allowing better coverage of the ocean at high latitudes; use of the FES2014 ocean tide model (instead of the GOT4.8) for all altimeter missions, providing a reduced sea level variance in many coastal and shelf areas, as well as at high latitudes (Fig. 8, left); use of the GPD+ wet troposphere correction (Fernandes and Lázaro 2016), leading to improved sea level variance estimate in coastal regions for most




altimeter missions compared to the previous non calibrated version (Fernandes et. al., 2015) and to other wet troposphere correction datasets (see Fig. 8, right for the example of the SARAL/AltiKa mission). Finally, the updated sea state bias correction used for some missions (such as Envisat) also contributes to better retrieval of mesoscale signals. The reader should refer to Quartly et al. (2017) for the details of the afore-mentioned corrections.

**4 Sea Level Budget Closure and Comparison with Model Outputs**

Different types of external validations of the SL_cci v2.0 products have been investigated. They are briefly described below.

**4.1 Global Mean Sea Level Budget Closure**

Closure of the global mean sea level (GMSL) budget implies that:

$$\Delta GMSL(t) = [\Delta MOcean\ (t) + \Delta SSL(t)] \quad (1),$$

where $\Delta$ means change of a given variable with time t; $\Delta MOcean\ (t)$ and $\Delta SSL(t)$ are time variable ocean mass and steric sea level components (SSL (t) being the depth integrated change in sea water density due to ocean temperature T and salinity S variations).

Water mass conservation in the climate system implies:

$$\Delta MOcean\ (t) = -[\ \Delta M\ (t)Glaciers(t) + \Delta MGreenland(t) + \Delta MAntarct.(t) + \ \Delta MLWS(t) + \Delta MAtmWV(t) + missing\ mass$$

$$terms\ \&\ errors]\quad (2),$$

where the $\Delta M(t)$ terms in the right hand side refer to glacier, Greenland and Antarctic mass balances, changes in land water storage (LWS) and atmospheric water vapor.

We have investigated to what extent Eq. (1) is verified using the SL_cci v2.0 GMSL and different data sets for the ocean mass and steric components. The various contributions to the GMSL are summed up to derive a 'synthetic' global mean sea

level. Consistency of the different products is evaluated and an error assessment study is performed. The synthetic global mean sea level is further compared to the global mean CCI sea level (both in terms of time series and trends).

We have considered the whole altimetry era (1993-2015) and used the various data sets considered in Dieng et al. (2017) to estimate the individual mass contributions (glaciers, ice sheets, land waters and atmospheric water vapor and snow) of Eq. (1). Fig. 9 shows the SL_cci v2.0 GMSL as well as the individual and sum of components over 1993-2015 (The 1.5 mm/yr

correction supposed to represent the TOPEX A drift has been applied to the GMSL time series for 1993-1998; see Dieng et al., 2017 for details).

We have also computed the residual time series (observed GMSL minus sum of components) over 1993-2015 using the SL_cci v2.0 time series as well as additional GMSL products provided by other groups (DUACS DT2014 distributed by CMEMS -previously AVISO-, University of Colorado, NOAA, NASA/GSFC,and CSIRO). Both residual trend and RMS

have been estimated. These are gathered in Table 1. We note that in terms of trend, all GMSL time series are very close, residual trends ranging from -0.11 mm/yr to +0.08 mm/yr. We conclude that looking at trend only does not allow us to



provide significant assessment the SL_cci v2.0 GMSL, although we note that the SL_cci v2.0 GMSL trend is the closest of the ensemble mean trend (difference of 0.02 mm/yr). The situation looks more favorable in terms of RMS, the SL_cci v2.0 time series giving the smallest RMS of the differences (Table 1 and Fig. 10). This confirms that no anomaly is present in the SL_cci v2.0 ECV and would suggest that it is better than other products at interannual time scales.

## 4.2 Comparison with ocean reanalyses

Assessment of the SL_cci v2.0 ECV has been carried out via multi-model approach, by comparing with Ocean ReAnalyses (ORA hereafter) from ECMWF. The reference ocean reanalysis product from ECMWF is the new ORAS5, which is closely related to the ORAP5 system (see Zuo et al., 2014 and Tietsche et al., 2015). ORAS5 is produced using NEMO Ocean Model coupled to LIM2 sea ice model (Barnier, et al., 2006). A series of observation types were assimilated in ORAS5 using NEMOVAR Ocean data assimilation system in its 3DVar FGAT (First Guess at Appropriate Time) approach. Observations assimilated in ORAS5 include ENsemble EN4 in-situ profiles, SLA from DUACS DT2014, SST from HadSST2 and Sea Ice Concentration from OSTIA. It is worth noting that radar altimetry SLAs were not assimilated in ORAS5 outside of 50°S to 50°N domain, or in any coastal region with bathymetry less than 500 m. Altimeter-derived GMSL variations were also assimilated for the satellite era using freshwater constraining in ORAS5 (See Zuo et al. 2015). A few other ORAs from ECMWF with slightly different configurations from ORAS5 (see Table 2) were also used here in order to estimate climate signal uncertainties. The gridded SLA maps (MSLA) from the SL_cci v2.0 ECV and ORAs were interpolated into the same regular 1x1° grid with an optimised land-sea-mask to facilitate inter-comparison. It is worth noting that the v1.1 ECV suffers from its imperfect land-sea mask, which makes non trivial the estimate of the global MSL and seasonal cycle signals. As mentioned earlier, an adequate land-sea mask has been used in ECV2.0.

Regional maps of MSL trend from SL_cci v2.0 ECV were evaluated against the DUACS DT2014 and ORAs, with the results shown in Fig. 11 (left panels). Thanks to the inclusion of two additional altimetry missions (CryoSat-2 and SARAL/AltiKa), ECV2.0 shows improved data coverage in the Arctic regions compared with ECV1.1, and more pronounced positive trend in the Beaufort Sea. This is consistent with ORAs and DUACS DT2014 results. This positive trend is visible both in ORAS5 and ORAS5-LW with relatively low uncertainty (±1.5 mm/yr), suggesting a robust climate signal in the western Arctic Ocean (see Giles et al., 2012). The spatial patterns of large uncertainties are reasonably consistent between SL_cci ECVs and ORAS5 (Fig. 11 (middle panels)), considering that these uncertainties were estimated following very different approaches. The sea level trend uncertainty in ECMWF ORAs is due to observation representativeness errors and forcing analysis errors in the ECMWF ocean data assimilation system (Zuo et al, 2017), while sea level trend errors from SL_cci are only associated with the formal error adjustment of the trends and are thus not representative of the total regional altimeter MSL trend uncertainty (see the section dedicated to the sea level uncertainties further in this paper for more details). Areas with large errors are normally associated with strong mesoscale eddy activities. Moderate sea level trend uncertainties (~1.2 mm/yr) were also observed in the Tropical Pacific and Southern Indian Ocean for ECV1.1 and ECV2.0. Compared with the SL_cci ECVs, ORAS5 is over-confident on its MSLA changes at most tropical



and subtropical regions, but less-confident in the Southern Ocean. An attribution study from ORAS5 reanalysis suggests that the mean sea level trend is dominated by the steric term while the mass variations are only important when considering the coastal regions (Fig. 11, right panels). The increase in the sea level in the Beaufort Gyre is almost entirely due to halo-steric changes (steric changes due to salinity variations), which is consistent with the changes of Arctic circulation in the Beaufort

Gyre (see the next paragraph) and the recent increase of freshwater there (Giles et al., 2012). The sharp front and reversing sea level trends signals in the North Atlantic suggests that the pathway of Gulf Stream extensions may be misrepresented in ORAS5, which is a common issue in ocean reanalysis.

**4.3 Comparison with the TOPAZ and NorESM models in the Arctic region**

The SL_cci v2.0 products for the high latitude seas and Arctic Ocean are also compared with and assessed against

complementary sea level fields derived from the TOPAZ data assimilation system and the Norwegian Earth System Model (NorESM) for the period 1993-2016. In the Sub-Polar Gyre the models and observations show smooth seasonal variability with comparable amplitudes of around 5-7 cm (see Figure 12, top panel). In addition a trend of just below +3 mm/yr is found for the observations and the NorESM simulation, while it is slightly less for the TOPAZ reanalysis. In the Lofoten Basin the comparison shows that the amplitude of seasonal signals are slightly larger (>10 cm), while the trend in the

SL_cci v2.0 observations is about 3 mm/yr compared with about 1.5-2 mm/yr for the model fields. These differences may result from the spatial pattern of the trend in the sea level rise which is more confined in the model fields than in the observations (see also Fig. 13).

In Fig. 13 the TOPAZ4 reanalyses fields are shown for the thermosteric and halosteric trends (upper) and the steric and total trends (lower). As noted in the ECMWF comparison (Fig. 11), the thermosteric trend in TOPAZ4 has little influence on the

steric and total trends in the Beaufort Gyre. In contrast the positive trend in the sea level rise appears to fully emerge from the halosteric trend except on the Siberian Shelf where it is most likely connected with the bottom pressure and hence related to water mass accumulation on the shelf. The apparent freshening of the Beaufort Gyre is consistent with findings reported by Morison et al. (2012) who proposed that this occurred as a result of persistent changes in the pathways of Arctic freshwater.

In comparison to the SL_cci v2.0 trends depicted in the Arctic Ocean, the TOPAZ4 sea surface height trends display more distinct regional structures of sea level rise and decline. A positive trend of about 6-7 mm/yr is found in the Beaufort Gyre but appears to extend over a very large area towards the Siberian Shelf which is not evident in the SL_cci v2.0 dataset. In the Nordic Seas and the Lofoten Basin, on the other hand, regions of both positive and negative trends stand out in contrast to the more gentle sea level rise expressed in the SL_cci v2.0 dataset. Moreover, the large and distinct region of strong decline

in the sea level of up to 3-4 mm/yr encountered in the North Atlantic is not found in the SL_cci v2.0 dataset which only give a very weak decline of around 1 mm/yr.

Looking at the Sub-Polar Gyre and the Lofoten Basin, on the other hand, one find that the key contribution to the positive trends in steric and total signals emerge from the thermosteric trends. This is assumed to be related to an increased





occupation by warm Atlantic water, which is further supported by the evidence of corresponding negative halosteric trends that would be expected provided the source is the warm and saline water emerging from the northwestward extension of the Gulf Stream into the North Atlantic Current.

The steric (thermo-, halo-) trends observed in the sea level in the Sub-Polar Gyre have also been discussed by Hatun et al., (2005). They presumed that the variable dynamics of the Sub-Polar Gyre controlled the respective inflows of either cold/fresh sub-polar waters from the East Greenland Current or warm/salty subtropical waters from the Gulf Stream and its extension into the North Atlantic Current (NAC). Using salinity criteria to identify the respective sources of the water masses, they showed opposing transport variability of both source waters. Evidently, this closely mimicked a strong SPG when the cold/fresh water transport is strong and vice versa a weak SPG circulation when the warm/saline water transports dominates the inflow to the gyre. In consistence with the findings presented here, it is therefore likely to conclude that a weakening of the anticlockwise circulation has occurred in the Sub-Polar Gyre during the last 20-25 years. In contrast, the distinct sea level rise encountered in the Beaufort Gyre during the same period has lead to an intensification of the clockwise circulation in the gyre that may stimulate more trapping of fresh and cold Arctic surface water.

The SL_cci v2.0 products have provided advanced opportunities for studies of sea level changes. The assessment of these products for the high latitude seas and the Arctic Ocean has focused on the Beaufort Gyre, the Sub-Polar Gyre and the Lofoten Basin in the Norwegian Sea. In so doing we have used the reanalyses from the TOPAZ4 operational system. The inter-comparison and assessment have documented interesting results and sometimes very good agreement and consistency between observations and models. In particular the findings and achievements include distinct evidence of sea level rise of around:

- 4-5 mm/year for the inner part of the Sub-Polar Gyre explained by the thermosteric contribution together with a barotropic source;
- 6-7 mm/year for the central Beaufort Gyre explained by the halosteric contribution and accumulation of fresh and cold Arctic water in the gyre;
- 3-4 mm/year in the inner part of the Lofoten Basin assumed to result from the thermosteric contribution resulting from increased residence time of Atlantic Water in the basin.

In consistence with these findings, it is furthermore likely to conclude that a weakening of the anticlockwise circulation has occurred in the Sub-Polar Gyre during the last 20-25 years, while the sea level rise in Beaufort Gyre during the same period has lead to an intensification of the clockwise circulation in the gyre with possible trapping of more freshwater.

## 4.4 Validation based on the GECCO model of the University of Hamburg

To assess the quality of the different SL_cci ECVs v1.1 and v2.0, the recent and higher resolution GECCO2 ocean synthesis framework (Köhl, 2014) has been used. The GECCO2 assimilation approach uses the adjoint method to adjust uncertain model parameters to bring the model into consistency with ocean observations. In this way, the ocean state estimation ultimately leads to new estimates of the surface forcing fields that are required to simulate the observed ocean in a best possible way (given the model resolution and the model physics). The GECCO2 solution covers the period from 1948 to



2011 and had been optimized over 23 iterations. See Köhl (2014) for a description of the GECCO2 ocean state estimate and the data sets used as constraints. Starting from this already optimized state, 2 additional assimilation runs (G0 and G1.1) were performed as part of this study, all starting from iteration 23, carrying out 5 additional iterations. The only difference between both assimilation runs being the different SSH data sets used as constraints, G0 assimilated the AVISO SSH fields

(SL0: DUACS DT2014 now distributed by CMEMS), whereas G1.1 assimilated the SSH fields from SL_cci v1.1 (SL1.1), see Scharffenberg et al. (2017).

Both daily mean GECCO2 synthesis results (G0 and G1.1) were interpolated onto the satellite tracks that matched the respective days for the respective along-track positions to be compared to DUACS DT2014 (SL0), SL_cci v1.1 (SL1.1) and to SL_cci v2.0 (SL2.0) satellite data sets. In order to compensate for the scales, the GECCO2 solution is able to resolve, the

satellite products SL0 and SL1.1 had to be filtered with an additional running mean filter of 9 points (f9), and SL2.0 with a running mean filter of 11 points (f11). The filter length was determined from the scales that GECCO2 manages to resolve in order to yield similar spectral characteristics of the respective signals, see Scharffenberg et al. (2017) for details. The model data comparisons have been performed separately for the ERS (ERS-1, ERS-2 and ENVISAT) and the TOPEX/Poseidon (TP) satellite series (TOPEX/Poseidon, Jason-1 and Jason2). Smaller model-data residuals suggest a better agreement

between the GECCO2 model and the satellite data sets.

Figure 14 shows the ratios of RMS difference based skill score as defined in Scharffenberg et al. (2017), for the TP (left) and the ERS time series (right). The total improvement due to the updated satellite data SL1.1 and its assimilation into the GECCO2 synthesis can be revealed by the ratio of the differences in G0 and SL0, by using only the DUACS DT2014 data set SL0, and of the differences in G1.1 and SL1.1 by using the previous updated SL1.1 data set only. This ratio

G0SL0/G1.1SL1.1 as shown in Figure 14 (top), highlights the reduction in the RMS differences for G1.1SL1.1 in most regions of the world oceans, leading to an improvement (red) of more than 30% in many regions. Hence, additionally to the equatorial regions, the Argentine shelf and parts of the ACC, improvements of more than 30% can be seen in the northern Indian Ocean, the north Pacific, subtropical regions, and large regions south of the ACC as well. Degradations of SL1.1 exist in isolated regions, where the GECCO2 synthesis adapts less well to the assimilated SL1.1 product. The regions showing a

degradation (blue) match with regions of small STD (see Scharffenberg et al., 2017), implying that the assumption of model serves as truth breaks down. The global mean (between 66°N and 66°S) improvement sums up to 4.75% and 4.74% for the TP and ERS data sets, respectively.

While the top panel gives the improvement from SL0 to SL1.1, the bottom panel answers the question about the total improvement from the DUACS DT2014 data set to the latest SL_cci v2.0 ECV. Here, the ratio G0SL0/G1.1SL2.0 compares

the different assimilation runs G0 and G1.1 while calculating the RMS differences to SL0 and the latest SL_cci ECV SL2.0. The improvement of SL_cci v2.0 ECV has now a more homogenous distribution. Only isolated regions have larger RMS differences for G1.1SL2.0, especially close to Antarctica as well as in the Arctic regions. The improvements for SL2.0 differ more between the TP and ERS data sets as it was the case for SL1.1. Especially in the equatorial regions the ERS data set has been improved. However, in most other parts of the world's ocean both satellite data sets see a clear improvement from





SL_cci v1.1 (SL1.1) to SL_cci v2.0 (SL2.0), especially in regions where SL1.1 did not improved much as compared to SL0. The overall global mean improvement from DUACS DT2014 to SL_cci v2.0 ECV sums up to 6.88% for the TP data set and 9.6% for the ERS data set. As the GECCO2 synthesis had assimilated SL1.1 but not SL2.0, the GECCO2 synthesis results G1.1 are not expected to be in best agreement to SL2.0.

Furthermore, the GECCO2 synthesis itself benefits from the assimilation of the SL1.1 product as well as has been shown in Scharffenberg et al. (2017). Thereby, the SL_cci v1.1 and 2.0 ECVs, generated by the ESA SL_cci project, have been improved significantly and are now in closer agreement with the GECCO2 synthesis and the various global oceanographic data sets assimilated therein (Köhl, 2014). For a detailed description and assessment of both SL_cci ECVs, we refer to Scharffenberg et al. (2017) and to the Climate Assessment Report (CAR, 2017).

## 10 4.5 Regional sea level validation: agreement with ocean model outputs

The gridded SL_cci v2.0 products have also been intercompared and assessed in the Mediterranean Sea against the sea level fields derived from three regional ocean models for the period 1993-2016. Over the shorter period 2002-2014 the assessment includes the comparison of the model steric field with the steric sea level derived from the combination of altimetric and GRACE gravimetric observations.

The two ocean simulations are the CNRM-RCM4 (Sevault et al., 2014) and the Protheus (Dell'Aquila et al., 2012), the ocean reanalysis is the CMEMS MEDSEA_REANALYSIS_PHIS_006_004, hereafter Med-MFC REA (http://marine.copernicus.eu). The global reanalysis ORAS5 is used as an additional comparison.

The first simulation CNRM-RCM4 is a fully coupled regional climate system model which includes a regional representation of the atmosphere, land surface, rivers and ocean. It is worth noting that the ocean NEMOMED8 model uses

the "Boussinesq" approximation (Mellor and Ezer, 1995) and the relaxation of the sea surface height in the Atlantic buffer zone. The same approximation is used in the Protheus model simulation. The regional reanalysis CMEMS Med-MFC REA assimilates the DUACS DT2014 SLA (now distributed by CMEMS, previously AVISO).

The model output elevation sea surface height (SSH) and steric components are compared to the observed sea level and to its steric component. The gridded SLA maps from the SL_cci v2.0 ECV and the elevation from the models were interpolated

into the same regular grid 0.25° x 0.25° to facilitate the intercomparison. The basin averages of the observed sea level, mass change and derived steric sea level component, and of the model sea level and steric components are displayed in Fig. 15.

For the models, which use the Boussinesq approximation, the total sea level is the sum of the model sea level and of the steric component basin average. At basin scales we find that the observed sea level agrees at best with the sum of thermo-steric and elevation basin average components. Results are shown for the CNRM model

in Fig. 16.

Regional maps of MSL trends from SL_cci v2.0 ECV were evaluated against the SSH from ocean models with the results shown in Figure 17 for the Protheus simulation and the CMEMS Med-MFC reanalysis. After the subtraction of the average trends of 2.5 mm/yr and 0.16 mm/yr from the regional ECV and CMEMS model maps (Figure 17, left, top and bottom), the



trend anomalies from observations and reanalysis show very similar spatial values (Figure 17, right, top and bottom). The regional maps of steric, thermo-steric and halo-steric trends are similar in the ocean simulations, differences between simulation and reanalysis are higher for the halo-steric component.

Basin averages of steric sea level from each model have been compared to the difference of measured total sea level and mass from GRACE. The seasonal amplitude of the model steric component is smaller than the satellite-derived steric component and phase is in good agreement. In Figure 18 annual basin averages of total sea level, mass-induced sea level and steric component, grouped at the top, middle and bottom of the figure respectively, obtained from observations and models are represented, for models the thermo-steric component is used. A significant correlation can be observed between the mass-induced sea surface heights measured by GRACE and both the observed altimeter SSH and the sum of both SSH and thermo-steric model component, see Fenoglio-Marc et al., 2012. The model and the steric sea level derived from altimeter and gravimetric observations show a similar long-term variability and some differences to be further investigated.

In summary, in the Mediterranean Sea the main differences both between the ocean model outputs and between the ocean model outputs and the observations are related to the halo-steric component, which trends have high negative values especially in the model reanalysis.

## 5 MSL Errors Characterization and Uncertainties

Major efforts have been carried out during the past few years to provide a user-oriented error budget of the altimeter sea level estimations. Such an error budget dedicated to the main temporal scales (long-term, interannual and seasonal signals) has been established by Ablain et al. (2015) and is given in Table 3. The GMSL trend uncertainty has been estimated to 0.5 mm/yr over the whole altimetry era (1993-2015) within a confidence interval of 90%. The associated sources of errors are related to some altimeter geophysical standards (Legeais et al., 2014, Couhert et al., 2015), the instabilities of the altimeter parameters (Ablain et al., 2012) and the multi-mission calibration (Zawadzki et al., 2016). Significant interannual variations are observed on the GMSL time series (Fig. 1) - mainly attributed to the ENSO (Ablain et al., 2017) - and contribute to the GMSL trend uncertainty in addition of all sources of errors described earlier (Cazenave et al., 2014). An uncertainty envelope for the GMSL has been proposed in order to better characterize inter-annual evolutions.

At regional scale, the sea level trends vary between +/- 5 mm/yr around the global mean of +3 mm/yr and the associated uncertainty is of the order of 2-3 mm/yr. A map of the regional MSL trends uncertainties has been estimated by Prandi et al. [2017], highlighting the regions where the trend estimations are higher than the level of uncertainty. At basin scale, two contributors to the altimeter trend uncertainty can be distinguished. The altimetry errors are one of the contributors. They can be related to the reduced quality of the altimeter sea level estimation in coastal areas and to the greater error of some geophysical altimeter corrections (ocean tide, inverse barometer and dynamic atmospheric corrections). The second contributor is related to the large internal variability of the observed ocean (and the fact that the associated trend may vary with the length of the record). The local variability is generated by regional changes in winds, pressure and ocean currents



which averaged out at global scale (e.g. Stammer et al. 2013) but this can significantly contribute to the sea level uncertainty at basin scale. One should note that for both global and regional sea level trends, uncertainties remain higher than the requirements of the GCOS [GCOS, 2011] of 0.3 mm/yr for the GMSL trend and 1 mm/yr for the regional MSL trend (see Table 3).

## 6 Conclusions and Perspectives

The ESA Climate Change Initiative has been the opportunity to realize the full potential of the long-term global Earth observations from satellite altimeters. This has led to the production of an accurate and stable sea level record designed to answer the user's needs for climate modelers and researchers. The quality assessment of this SL_cci v2.0 ECV has been carried out distinguishing different temporal and spatial wavelengths and following different approaches: comparisons to the previous version of the ECV and to altimeter products from other groups, sea level budget closure approach and comparison with model outputs.

Compared to the previous v1.1 version of the SL_cci ECV (Ablain et al., 2015), the main observed differences are related to the updated v2.0 altimeter standards that have been selected within the SL_cci phase II project and used to calculate the altimeter sea level anomalies (see Quartly et al., 2017 for more details on these standards). One of the major differences between both versions is associated to the increased number of altimeters available in the satellite constellation used for the SL_cci v2.0 ECV, as compared to the v1.1 ECV. This has led to an improved sea level variance in the reprocessed ECV thanks to the improved sampling of the ocean at the end of the period. This highlights the importance for climate products of using a minimum of two satellites in the sea level ECV production and also to ensure that the number of such satellites remains stable in the constellation.

The different reference time periods used in both versions of the ECV is related to the different Mean Sea Surfaces used to compute the Sea Level Anomalies : DTU10 in v1.1 is referenced to 1993-2008 whereas DTU15 in v2.0 is referenced to 1993-2012. This has to be taken into account in the context of data assimilation for ocean models.

At global scale, the v2.0 sea level trend is the same as in the v1.1 ECV when considering the total altimeter period. However, the use of the new GNSS Path delay (GPD+) wet troposphere correction (Fernandes and Lázaro, 2016) significantly affects the trend at decadal time scale (up to 0.2 mm/yr for each altimeter decade). This is of major importance for sea level budget closure studies which usually focus on the 2015 onwards period. At regional scale, up to +/- 1 mm/yr sea level trend differences are observed compared to the previous version of the ECV and the large scale differences are associated with the updated orbit solutions used in the v2.0 ECV (Quartly et al., 2017).

Regarding the annual cycle of the sea level, a small difference of amplitude is observed between SL_cci v1.1 and v2.0. Comparisons with the in-situ measurements from tide gauges and from the combination of the dynamic heights derived from

temperature and salinity profiles of Argo floats and the GRACE ocean mass contribution indicate that the SL_cci reprocessed ECV is slightly closer to the in-situ reference.

The comparison with sea level time series from other altimetry groups and budget closure studies have demonstrated the high quality of the reprocessed SL_cci sea level record.

During the project, the altimetry measurements errors and associated uncertainties have been better estimated separating the main temporal and spatial scales (Ablain et al., 2015). An estimation of sea level uncertainties has highlighted that in some regions, errors are greater than the signal itself. This work will significantly contribute to increase the accuracy of climate studies. It is worth noting that in spite of the improved altimeter standards used in the product, the GCOS user requirements (GCOS, 2011) are still not reached considering some specific scales (e.g. 0.5 mm/yr uncertainty for the GMSL trend

compared to the 0.3 mm/yr requirement).

The reprocessed SL_cci v2.0 ECV is thus the state of the art sea level ECV available for climate studies. Following the end of the ESA SL_cci project in 2017, the operational production of the sea level ECV has been transferred to the European Copernicus Climate Change Service (C3S) which will set up the routine and sustained production of the ECV. However, a

strong need to continue research and development for the sea level record has been identified. Perspectives of evolution include the improvement of the sea level estimation in coastal areas and in ice-covered regions, the better characterization of the sea level uncertainties and the quality improvement of the altimeter observations. This will contribute to improve the quality of the sea level ECV and come closer to the GCOS requirements (GCOS, 2016).

**Author contribution**

Phase 2 of the Sea Level CCI project was managed by JFL, who oversaw the production and validation of the SL_cci ECV v2.0. The initial draft of the paper was written by JFL with the contributions of GQ for Sect. 2, AC for Sect. 4.1, HZ for Sect. 4.2, JJ for Sect. 4.3, MS for Sect. 4.4 and LFM for Sect. 4.5. Other authors contributed through their revision of the text.

**Competing interests**

The authors declare that they have no conflict of interest.

**Acknowledgments**

The authors acknowledge the support of ESA in the frame of the Sea Level CCI project, launched and co-ordinated by technical officer Jérôme Benveniste. It was also made possible thanks to the support of CNES for several years with the use of the DUACS altimeter processing system. We would also like to thank all contributors to this project who have



participated actively in the SL_cci project, with special recognition to A. Ambrozio and M. Restano in support of ESA, for their diligent reviewing of all the documents and data sets produced by the SL_cci team.

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

ECMWF Tech Memo (795)



|  | Trends (mm/yr) | RMS (mm) |
|---|---|---|
|  | Jan 1993 - Dec 2015 | Jan 1993 - Dec 2015 |
| AVISO - Ensemble mean GMSL | -0.07 | 0.77 |
| CU - Ensemble mean GMSL | 0.07 | 1.23 |
| NOAA - Ensemble mean GMSL | -0.11 | 1.13 |
| NASA - Ensemble mean GMSL | 0.08 | 0.91 |
| CSIRO - Ensemble mean GMSL | 0.02 | 1.22 |
| CCI (V2.0) - Ensemble mean GMSL | 0.02 | 0.69 |

**Table 1: Trend and root mean squares of the difference between individual GMSL time series and ensemble mean GMSL.**

| Description | Resolution | Assimilation | Period |
|---|---|---|---|
| DUACS DT2014 | 0.25°x0.25° | - | 1993-2015 |
| SL_cci ECV1.1 | 0.25°x0.25° | - | 1993-2014 |
| SL_cci ECV2.0 | 0.25°x0.25° | - | 1993-2015 |
| ORAS5 | 0.25°x0.25° | SST, SIC, T, S, SLA | 1993-2015 |
| ORAS5-LW | 1°x1° | SST, SIC, T, S, SLA | 1975-2015 |

**Table 2: Summary of the ORAs used for the SL_cci v2.0 ECV evaluation. DUACS DT2014 MSLA (Pujol et al., 2016) is now**

5 **distributed by CMEMS (previously AVISO). ORAS5 is the ECMWF quarter degree resolution ocean-sea ice reanalysis; ORAS5-LW is the ORAS5-equivalent low resolution (ORCA1 grid, with approximately 1° resolution, with meridional refinement at the Equator) reanalysis. Both ORAS5 and ORAS5-LW have five ensemble members, generated by a generic perturbation scheme (Zuo et al, 2017).**

| Spatial Scales | Temporal Scales | Altimetry uncertainties | User requirements |
|---|---|---|---|
| Global MSL | Long-term evolution (> 10 years) | < 0.5 mm/yr | 0.3 mm/yr |
|  | Interannual signals (< 5 years) | < 2 mm over 1 year | 0.5 mm over 1 year |
|  | Annual signals | < 1 mm | Not defined |
| Regional sea-level | Long-term evolution (> 10 years) | < 3 mm/yr | 1 mm/yr |
|  | Annual signals | < 1 cm | Not defined |

10 **Table 3: Mean seal level error budget for the main climate scales [Ablain et al, 2015]**



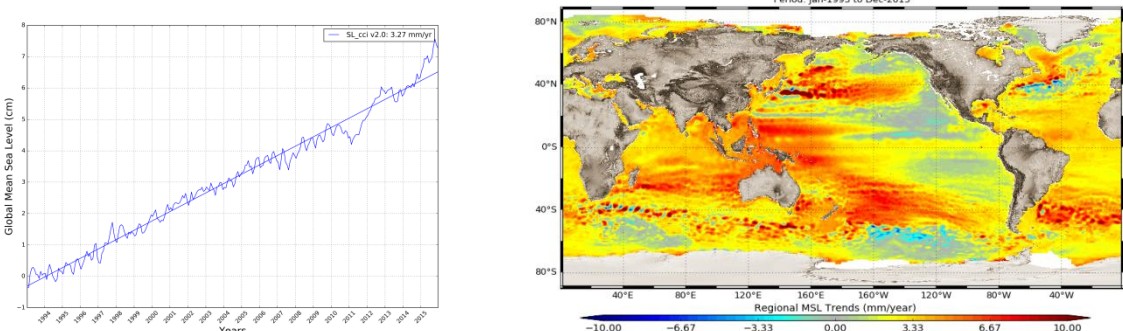

**Figure 1: Global (left) and regional (right) MSL trend (from SL_cci v2.0 ECV). The GMSL has been adjusted from annual and semi-annual signals.**

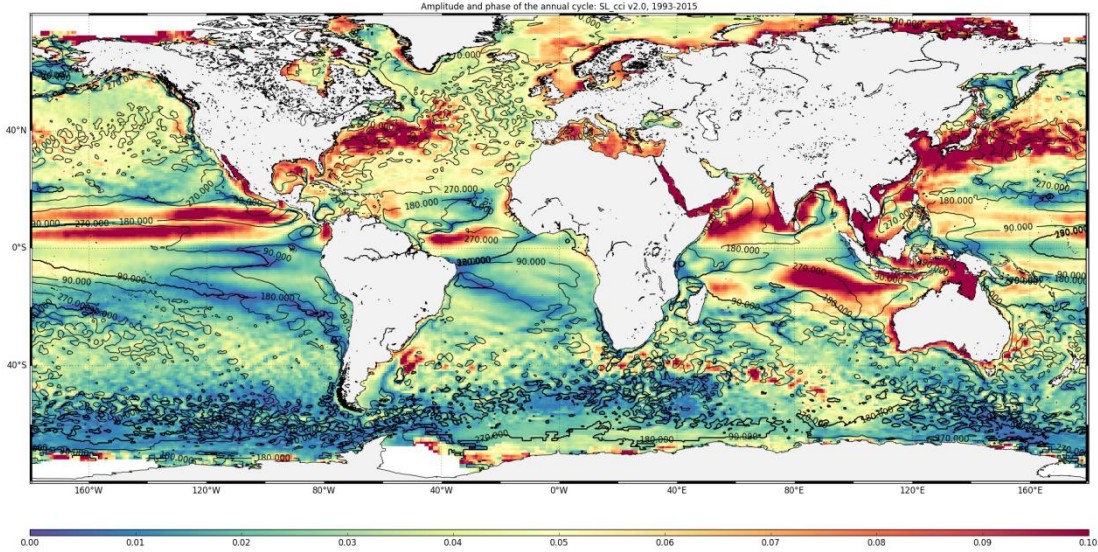

**Figure 2: Global Amplitude (colored contours, between 0 and 0.1 m) and phase (superimposed black isolines, between 0 and 360°) of the annual cycle of the SL_cci ECV v2.0 during 1993-2015.**



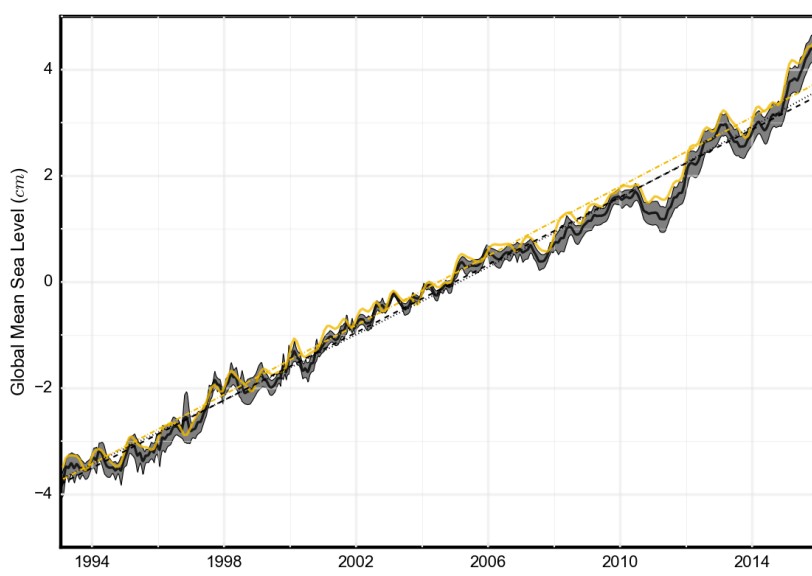

**Figure 3: Comparison of the SL_cci v2.0 global MSL (in yellow) and the ensemble mean of the global MSL derived from different groups (DUACS DT2014, CSIRO, Colorado University, GSFC and NOAA) during the period 1993-2015. The seasonal variations have been removed.**

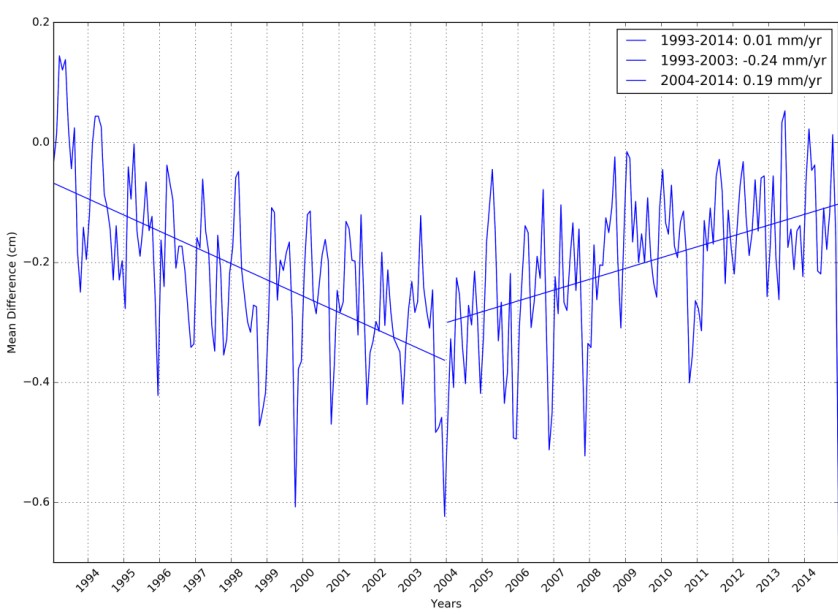

**Figure 4: Global mean sea level differences between the SL_cci ECV v2.0 and v1.1. The trends are indicated for the total record (1993-2014), for 1993-2003 and 2004-2014. A jump is observed in mid-2008, illustrating the corrected anomaly in ECV v2.0.**





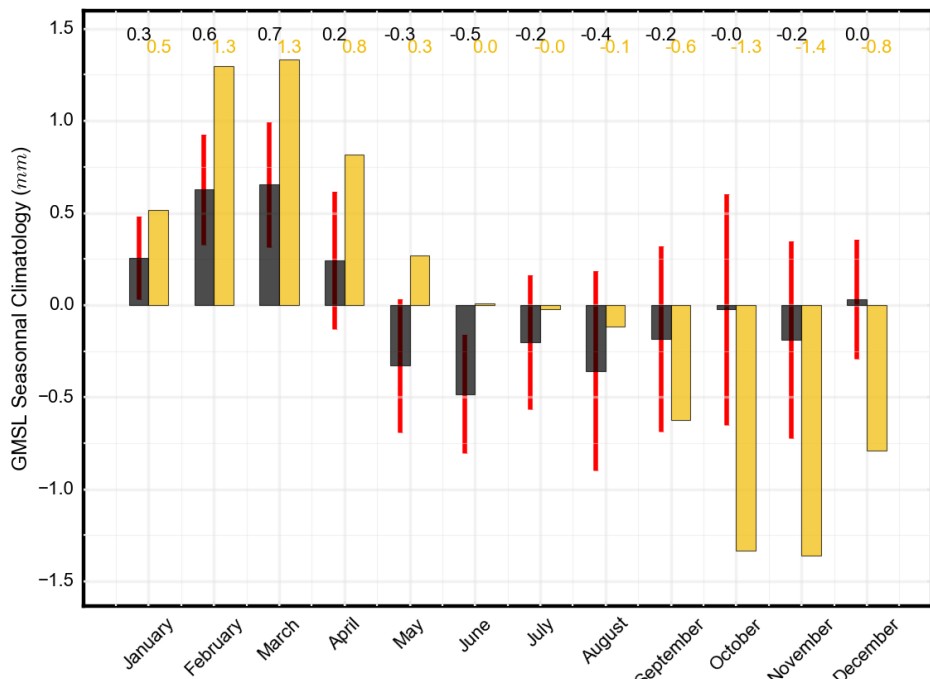

**Figure 5: SL_cci v2.0 global MSL monthly climatology (yellow) compared to the ensemble mean (black) of the monthly climatology derived from different groups (DUACS DT2014, CSIRO, Colorado University, GSFC and NOAA) and the red bars show the associated standard deviation. The period considered for the monthly average is 1993-2015.**

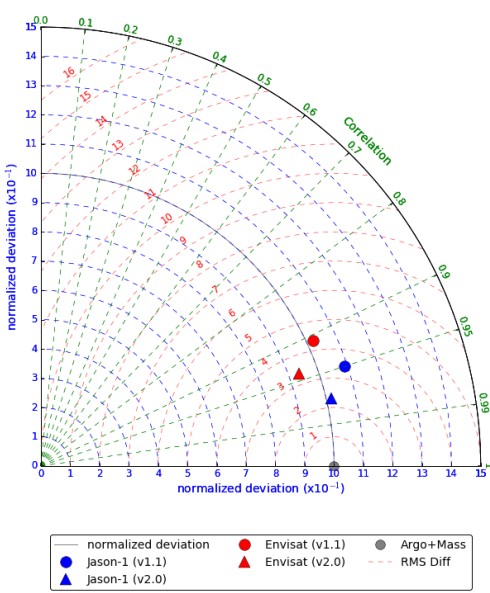

**Figure 6: Taylor diagram of the annual signal of the Envisat (in red) and Jason-1 (in blue) sea level anomalies (2005-2012) calculated considering the pole tide corrections derived both by Wahr 1985 (circle, used in ECV v1.1) and by Desai 2015 (triangle). They are compared with the independent sea level estimation (grey dot) derived from the in-situ Argo dynamic heights anomalies (referenced to 900 dbar) and the GRACE ocean mass contribution (GRGS RL03v1).**

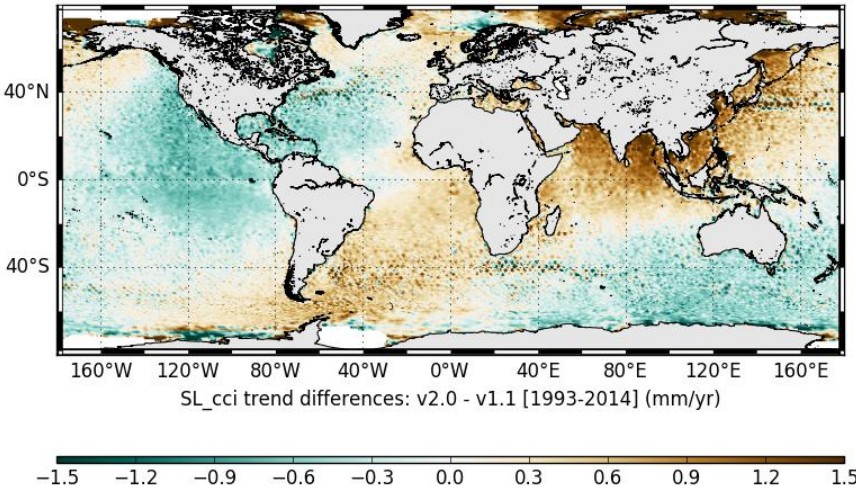

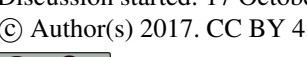

**Figure 7: Regional mean sea level trends differences between the SL_cci ECV v2.0 and v1.1 during 1993-2014.**

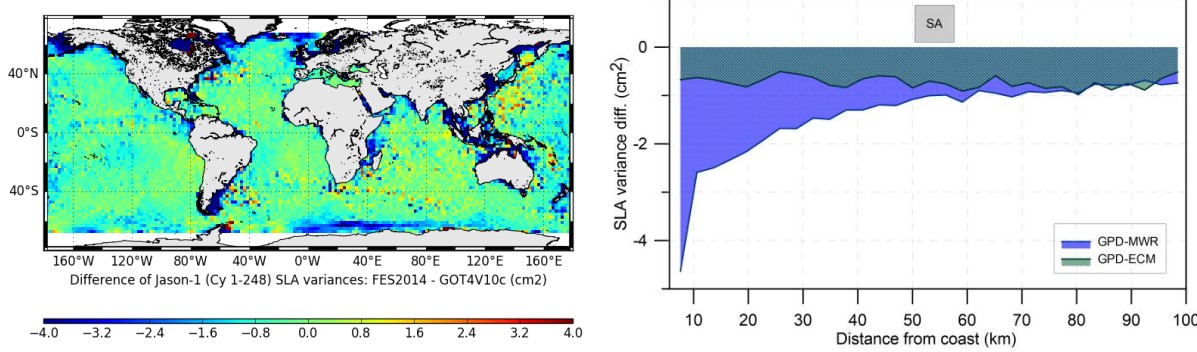

**Figure 8: Left: sea level variance differences for Jason-1 cycles 1-248 (2002-2008) using successively FES2014 and GOT4.10c ocean tide corrections. Right: SARAL/AltiKa sea level variance difference calculated with different wet troposphere corrections (green compares GPD+ with ECMWF operational model and purple with the initial version of the radiometer correction) as a function of the coastal distance.**

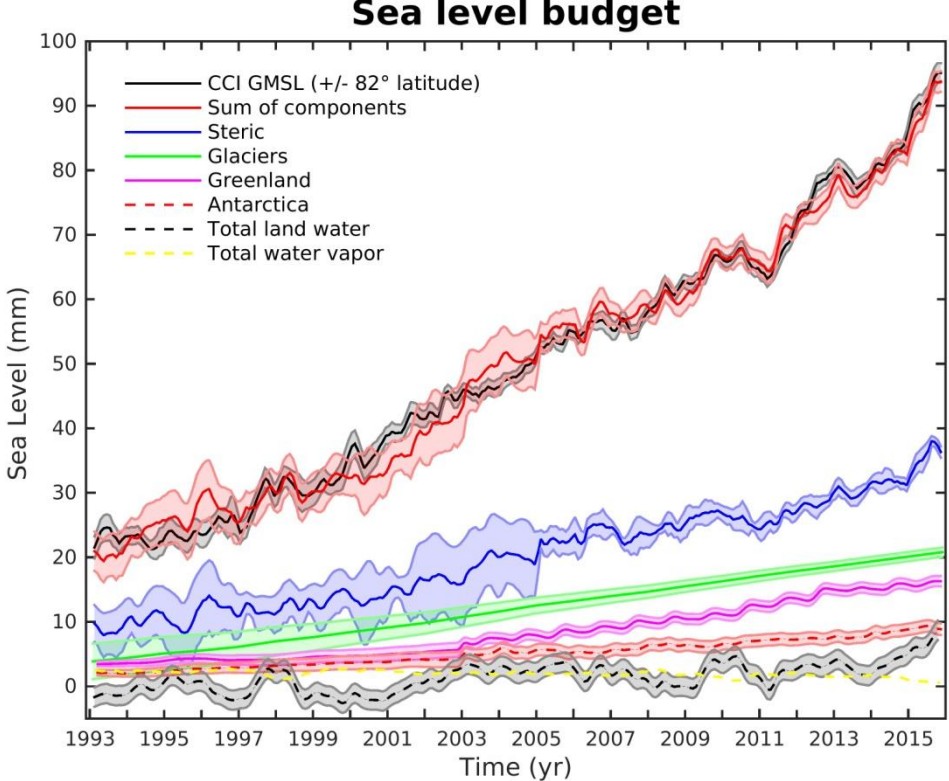

**Figure 9: Sea level budget using the SL_cci v2.0 GMSL time series (adapted from Dieng et al., 2017).**




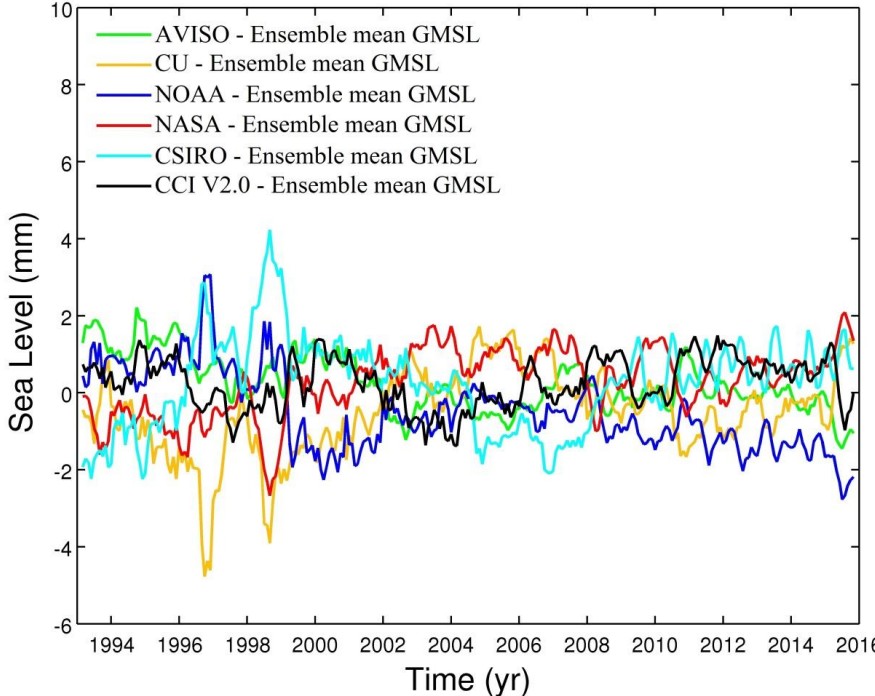

**Figure 10: Differences between individual GMSL time series and ensemble mean GMSL (average of the 6 products) over January 1993-December 2015 (adapted from Dieng et al., 2017).**



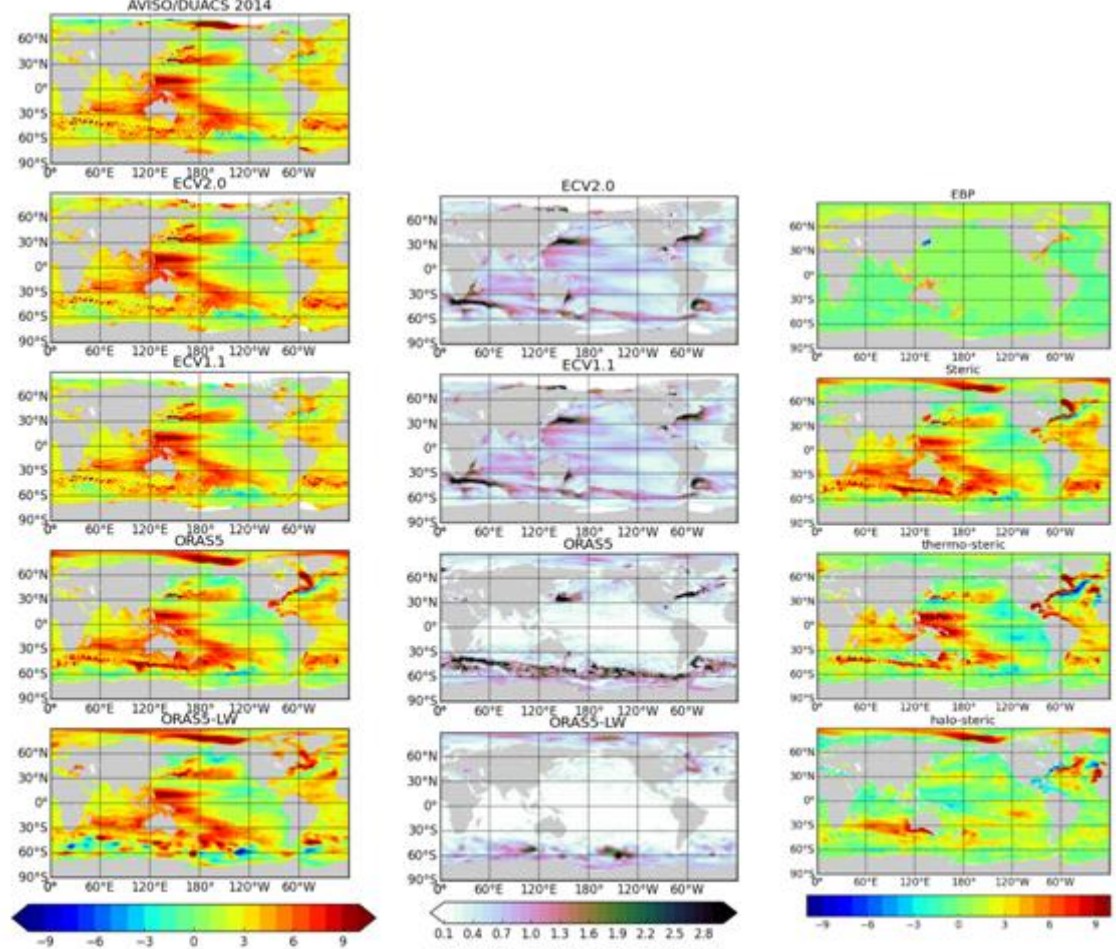

**Figure 11: (left panels) Mean sea-level trends (in mm/yr) and (middle panels) uncertainties in DUACS DT2014, SL_cci ECV v2.0 and v1.1, ORAS5 and ORAS5-LW. MSL trends are computed using monthly mean sea level data from 1993-2014. Trend uncertainties have been provided as climate indicator with ECV products, and were estimated using ensemble spread from 5 ensemble members of ECMWF ORAs; (right panels) Attributions of MSL trends derived from ORAS5 as, from top to bottom, Equivalent Bottom Pressure (EBP) mass variations, steric changes, thermo-steric and halo-steric changes for the same period.**



**Figure 12:** Seasonal to annual changes in sea level in mm/yr for the period 1993-2016 for the Sub-Polar Gyre (upper) and the Lofoten Basin (lower). For the Lofoten Basin the DTU-based sea level change is also displayed for the comparison. For both plots the vertical axis is in cm.



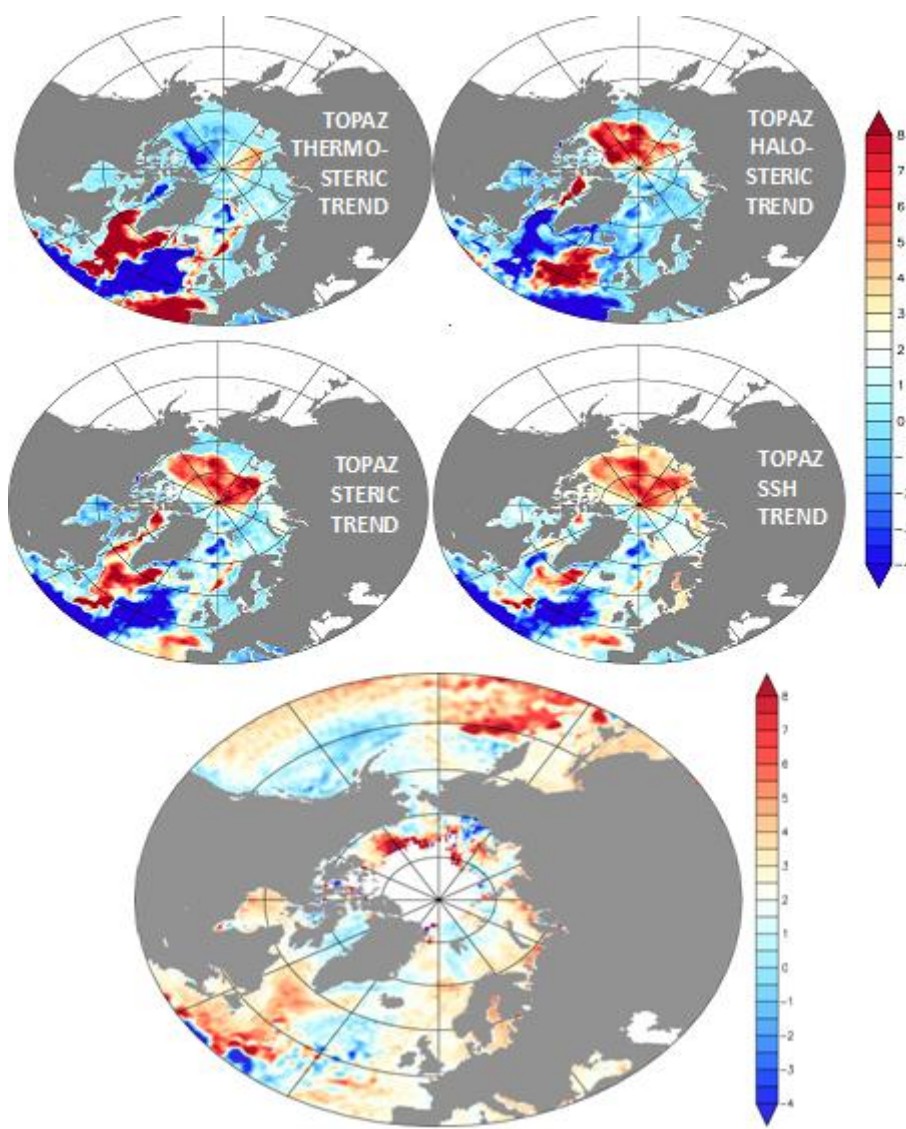

**Figure 13: Contribution to the sea level trend (mm/yr) from TOPAZ4 reanalyses for the period 1993-2016 for (top left) the thermosteric contribution, (top right) the halosteric contribution, (middle left) the total steric trend, and (middle right) the total trend. The observed trend from the ESA SL_cci v2.0 data (bottom).**



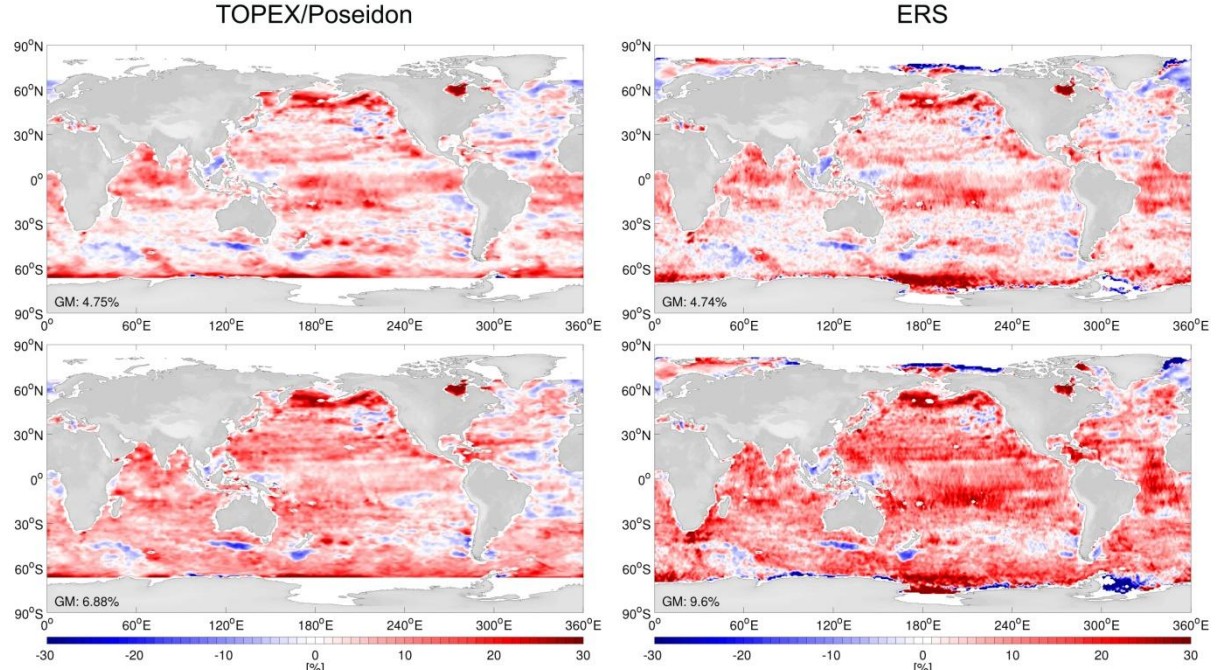

**Figure 14: Ratio of RMS differences in low-pass (f9/f11)-filtered data, for (top) the improvement from DUACS DT2014 (now distributed by CMEMS, previously AVISO) to SL_cci v1.1 as G0SL0/G1.1SL1.1, and (bottom) the total improvement from DUACS DT2014 to SL_cci v2.0 as G0SL0/G1.1SL2.0, for TP time series left, and for ERS time series right.**

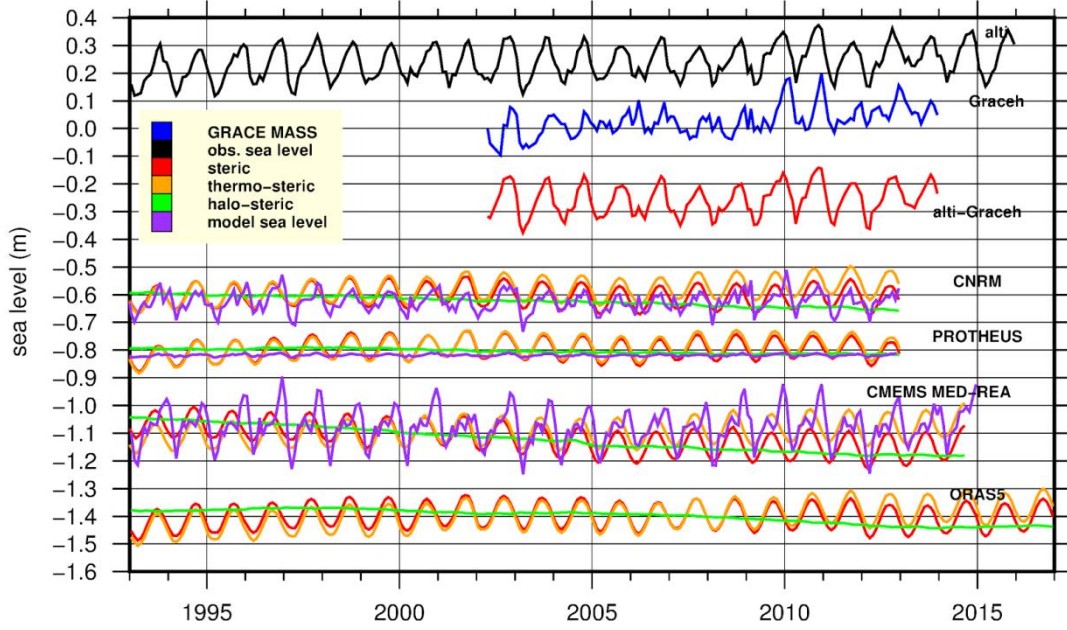

**Figure 15: From top to bottom, basin average of altimeter sea level, GRACE mass change, steric derived from altimeter sea level and mass component, steric and sea level components of two simulations and a reanalysis and of a global model.**





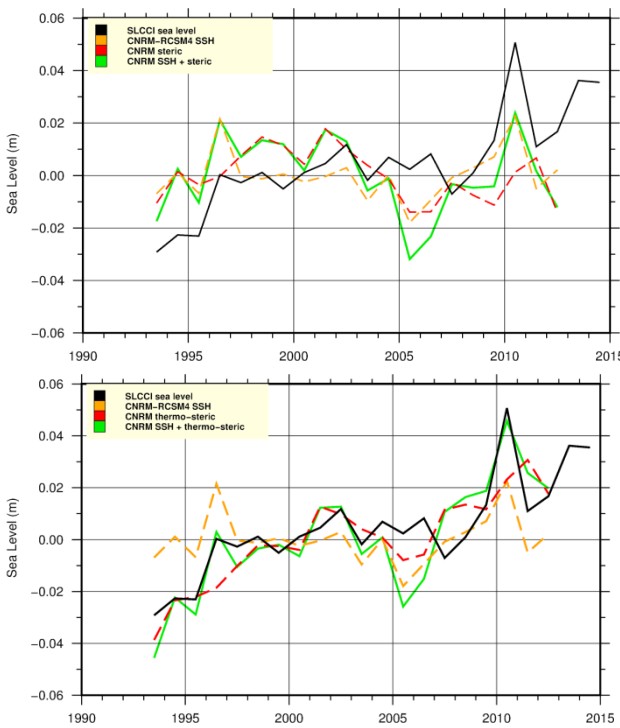

**Figure 16: CNRM versus SL_cci total sea level. Top: Sea surface height (green) from elevation plus steric compared to SL_cci (black), with elevation (orange) and steric (red) components. Bottom: as in top with thermo-steric instead of steric.**



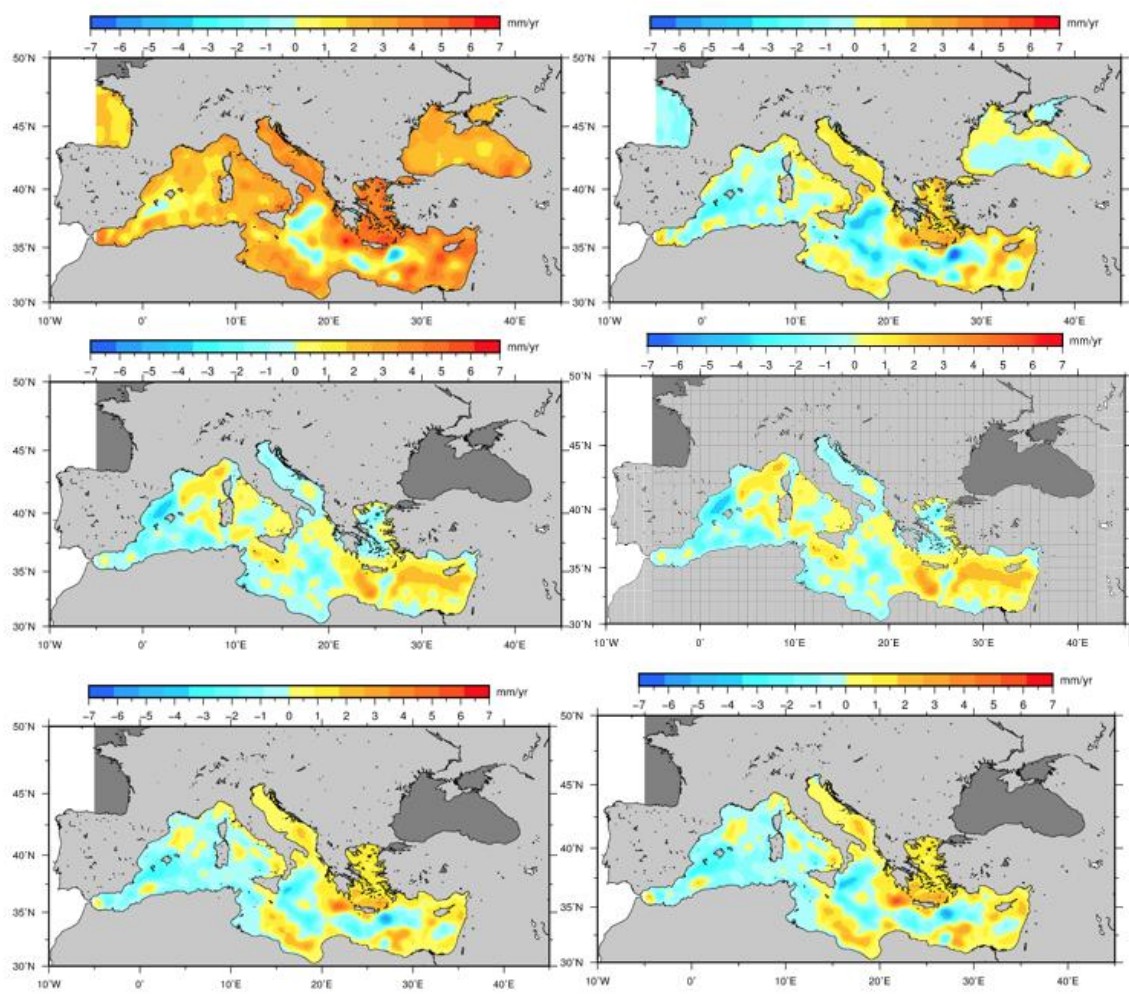

**Figure 17: Trend (left) and trend anomalies (right) of SSH from SL_cci (top), Protheus model simulation (middle) and CMEMS reanalysis (bottom).**

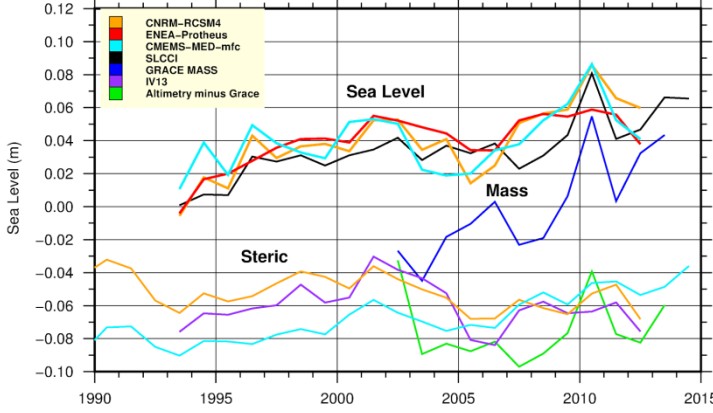



**Figure 18: Sea level (top), mass (middle) and steric (bottom) components from SL_cci and GRACE observations and from models. The color indicates the source, the location in the figure (top, middle or bottom) indicates the type of product (sea level or steric component for the model output)**