# Peer review of "An Accurate and Homogeneous Altimeter Sea Level Record from the ESA Climate Change Initiative"

_Earth System Science Data, 2017_

## Referee Comment (RC1) · Anonymous Referee #1 · 8 Nov 2017

In this paper, the authors illustrate an altimeter Sea Level Record generated within the ESA Climate Change Initiative.

The article fits properly with the aim and scope of the journal. The data set described in the article is certainly unique and useful for ocean sea level monitoring and understanding of its variability as well as the factors at the origin of observed changes. Moreover, the data set is also important to validate climate models used for projecting future changes. In summary, an accurate and homogeneous altimeter sea level record from multi-mission altimetry is very welcome in the ocean and climate communities. The data set is in netcdf and usable in its current format and size. The metadata are

appropriate and everything is well documented in reports and published papers.

With reference to the content of the article, the authors describe the last reprocessed product and then provide some validation results obtained through different approaches. The author use other sea level data sets provided by other groups. Comparisons are made with tide gauge measurements, other sea level data sets generated using available source (Argo and GRACE) as well as model outputs. The sea level errors and uncertainties are discussed and the perspectives of evolution of the sea level product are provided.

This paper is well written and clear, structured properly to support the publication of the data set. The material is also presented in very good editing style. All figures are carefully prepared and clearly illustrated. The language is consistent and only few typos are identified. The rationale is explained clearly. The methods and materials are described in sufficient detail in previous published papers that are properly cited. The comparisons are carried out with rigour. The conclusions and perspectives are included in an appropriate way at end of paper.

Nevertheless, I would like to draw attention of the authors to some important points:

1) Sources of errors are discussed in the article as well as discrepancies, however, a reader is left without an error quantification for some outputs of the data set. I would expect error bars for global sea level rise and error maps for regional trend map. Moreover, I don't see any reference to the IPCC, where sea level was well considered, e.g. will this data set contribute to the next IPCC report? 2) Two new Arctic sea level records are mentioned and cited but nothing is showed in the article. Due to the importance of the Arctic Sea, some results have to be presented and discussed, in particular agreements/disagreements between the two products and some validation and comparison of the two products. Why the first data set does not include ERS and CryoSat? 3) The data set accessible via the given DOI identifier, however, the access mechanism to data is questionable as people have to send an e-mail and get permission. I see

the producers want to take track of users , but there are automatic methods to do that, while users want download easy and immediate.

This is a paper that deserves publishing in ESSD after above points are addressed.

What follows are some typos identified:

Pg. 4, row 23, "Data from the other missions (also called complementary missions) contribute to improve….": add "that" before contribute the make the whole sentence correct

P. 5, row 16, "are", change to "is"

Pg. 13, row 27-29, please apply right character type and sie

Pg 19, row 31, "Passaro, M.,": update the reference (in press or published)

Pg. 20, row 4, "Prandi ..": update the reference (in press or published)

―――――――――――――――――

---

## Referee Comment (RC2) · Anonymous Referee #2 · 16 Nov 2017

The aim of the paper is well addressed, which focuses on the review of the SL_cci v2.0 ECV and the associated uncertainty, as well as the validation process. Comparison between the updated v2 product with the previous v1 and other datasets (from different groups) is also discussed, making the study relevant to various readers.

In general, the article is well written and clear. The structure and flow of the paper are in appropriate order.

Only some minor points:

Pg 2. Row 4. in the Abstract, authors mentioned about the GCOS requirements. But no clear explanation is given in the Abstract nor in the Introduction. The reader is left

without clear explanation about what actually are the GCOS requirements.

Pg 6. Row 10. Author wrote 'In the v1.1 SL_cci ECV, a 1 mm jump was found in the GMSL around mid 2008'. Is this referred to Fig.4? If so, please indicate in the sentence.

Pg 7. Row 26. +3 cm2. 2 should be written in superscript.

Pg 24 Figure 3. Does the x-axis referred to Years? If so, please indicate. The legend should be improved. What are the solid and dashed lines in black (and dashed lines in yellow)?

Pg 24 Figure 4. In the Figure caption and text, Author highlights a jumped about 1mm in mid-2008. Unfortunately, the jump is not obviously appeared in the Figure. I suggest the author highlights the jump area by showing it in a scar-box.

---

## Referee Comment (RC3) · Anonymous Referee #3 · 5 Dec 2017

Journal: Earth System Science Data Discussions

Title: An Accurate and Homogeneous Altimeter Sea Level Record from the ESA Climate Change Initiative

Author(s): Jean-Francois Legeais et al.

MS No.: essd-2017-116

General Comments:

This paper presents the ESA CCI v2.0 sea level dataset, as introduced in detail by Quartly et al. (2017). The dataset described is an advance from the initial v1.1 release,

and represents a significant contribution to the user community. The data has unique components, is largely complete and is considered to be useful for a wide user group. The paper is generally well structured and well presented. I have identified a number of issues that require clarification – these largely concern how some data quality issues are articulated and presented within the manuscript. I have also identified a number of minor corrections that should be considered. With this in mind, I consider the work to be within scope of ESSD and worthy of publication pending consideration of the issues articulated below.

Major Issues:

1) There is somewhat of a contradiction that flows through this manuscript, and remains present into the conclusions. The work claims to present an accurate GMSL record, yet awkwardly introduces the systematic bias associated with TOPEX-A, as brought to attention by Watson et al. Nat. Clim. Chg. (2015). The work also cites Valladeau et al (2012), Dieng et al (2017) and Beckley et al (2017) around this issue, noting the omission of a citation of the relevant Chen et al., Nat. Clim. Chg. (2017) work.

As the authors correctly highlight, these recent advances have led to a downward systematic revision of the rate of change in GMSL (that isn't accounted for in the CCI v2.0 dataset). Given the news piece by Tollefson (Nature News, 2017) and subsequent letters to the editor by Nerem, Cazenave and Church (Nature, 2017), it is important to avoid confusion on this issue.

This manuscript describes a dataset that *does not* correct for the TOPEX related issues, yet parts of its accuracy assessment (e.g. assessment of the sea level budget, Figure 9) *does* correct for these issues (Pg8, Line 25), thus presenting a somewhat misleading perspective. The authors also present trend differences (record minus ensemble mean) as a measure of how robust the CCI record is (Table 1) – this is somewhat misleading as all of these records contain the error associated with the TOPEX record. I'm not convinced the related uncertainty section is robust without at least some

discussion of issues associated with TOPEX (in particular, the A/B bias, retracking etc)

I contend that these issues need to be made entirely clear is a revised manuscript. The conclusions need to include some remarks on this important issue, pointing towards the future release of a reprocessed TOPEX dataset. I would also recommend revising the title to "An improved and homogeneous...".

2) The section on regional rate uncertainties is presently poorly evidenced. The manuscript cites work that is "in prep", and no real mention of meaningful uncertainties around regional trends are provided. This should be addressed in a revised version of the manuscript.

Mostly Minor Issues (P=page#, L=line#, suggested rewording is given between dots "..."):

P2 L4: Reference required after GCOS requirements

P2 L7: Global Mean Sea Level is abbreviated here as MSL but later as GMSL

...heat added to the ocean...

P2 L8: ...monitoring of sea level...

P2 L10: validation of climate models

P2 L11: "supposed to correctly" isn't this implicit? I recommend removing.

...altimetry mission have delivered...

P2 L13: Suggesting adding e.g. before these references as this is not exhaustive

P2 L17: ...This program aims to realise the full...

P2 L20: ...distributed to the user community.

P3 L5: I don't think these brackets are required.

P3 L11: The sentence beginning Thus a single... doesn't make sense and needs to

be reworded.

P3 L16: The early section defines v2.0 incorporating 2014-2017 data. This needs to be clarified – it is really 2014-2016 inclusive? If so, does it include Jason-3. This needs clarification.

P3 L19: ..since sea level estimation from altimetry requires. . .

P3 L21: Again, insert e.g. before references.

P3 L23: . . .most appropriate solution/algorithm to ensure. . .

P3 L26: remove the word analysis

P3 L31: remove quotes from around pole tide

P4 L11: . . .SL_cci products (including the SLA) (reference). . .

P4 L12: . . .both datasets should be. . .

P4 L31: . . .for users. The . . .

P5 L8: . . .has been carried out over different spatial and temporal scales.

P5 L12: Here the CI is given as 95% but on page 14 it is 90%. Clarify.

P5 L12-25: See my significant remarks around how this section is articulated (and should be improved) in the top section of my review.

P5 L16: . . .However, over decadal time scales, the v2.0 GMSL trends are significantly different to those from v1.1. . .

P5 L25: I notice here that Chen et al. Nature Climate Change, 2017 is not referenced, and should be.

P5 L25: After mentioning the 1.5 mm/yr in terms of the GMSL trend, the date period for this metric should be provided.

P5 L28: GMSL rise is not accelerating, GMSL is accelerating. Rise is typically thought of as a rate of change. That rate is increasing, i.e. the level is accelerating. You have one too many derivatives here.

P6 L6: ...ocean models. Users interested...

P6 L11: ...correction, and present in the Radar... (This sentence is unclear, why is RADS introduced here? Clarify).

P6 L20: ...smoother periodic signal... (it is unlikely to be a pure sinusoid)

P6 L21: Figure 5 – is this correct? I'm surprised by the lack of agreement in the annual terms from other labs. Please confirm this has been computed correctly using the same smoothing across all data centres.

P6 L26: ...sea level difference computed against in-site data...

P6 L32: What independent data? Clarify.

P7 L4: ...both datasets...

P7 L9: ...Southern Ocean, the...

P7 L16: Are these differences significant given the difference in data span? See my remarks around how regional uncertainties are addressed in the top section of my review.

P7 L32: ...variance estimated...

P8 L14: The glacier mass term in the equation has an additional (t) in it that needs to be removed.

P8 L17: ...refer to changes in mean glacier, Greenland and Antarctic mass balances, land water storage (LWS) and atmospheric water vapor (AtmWV).

P8 L25: See my significant remarks around how this section is articulated (and should be improved) in the top section of my review.

P8 L30: The use of "ensemble mean" is slightly unclear. . . in this section, is the ensemble mean referring to the sum of the terms in the budget? This needs to be clarified in captions and text.

P8 L31: . . .that looking at solely the trend does not allow. . .

P9 L1: This statement is somewhat disingenuous – it is infact equal closest as the CSIRO dataset has the same difference.

P9 L3: Regarding Fig 10, around 1996-1999 it appears CU and CSIRO are quite different to the ensemble mean, but CCI is not. Interestingly, CU and CSIRO seem anti-correlated. This doesn't appear to be mentioned in the manuscript but is likely an important point worthy of a remark.

P9 L4: This sentence could be improved – "no anomaly" is not exactly the case (refer to the TOPEX issue, see comments at the top of my review) and "better" is a poor choice of words, consider revising.

P9 L8: . . .using the NEMO. . .

P9 L14: "freshwater constraining" is constraining the correct word here?

P9 L18: . . .which makes the estimation of GMSL (and its seasonal cycles) non trivial.

P9 L26: . . .(middle panels, Fig. 11). . .

P10 L25-30: This discussion does not mention regional uncertainty at all.

P11 L10: . . . gyre. Consistent with the findings presented here, it is possible to. . ..

P11: Sometimes SPG is abbreviated, sometimes it isn't, be consistent.

P11 L14: Is advanced the best word here? Perhaps new?

P11 L26: This repeats line 14. Revise.

P12 L34: datasets (no space)
P13 L1: ...improve much compared...

P13 L2: one dec place to be consistent

P13 L27-30: Font issue here

P14 L6: ...In Figure 18, annual...

P14 L8: ...represented (for model datasets, the thermos-steric component is used).

P14 L10: Put see reference in brackets.

P14 L13: ...which have high negative trend values, especially... The especially part here is ambiguous as the sentence refers to differences between models, and then between modes and obs. Suggest reword this sentence.

P14 L15: Error not Errors

P14 L17: estimation not estimations

P14 L19: 90% or 95% as earlier indicated?

P14 L25: This section is weak – there is no robust discussion around regional uncertainty. The reference is "in prep". If this is published, update, else remove and reword this section. There should be discussion around the noise model used to estimate "order of 2-3 mm/yr", as I'm sure that isn't a 90% CI value when an appropriate noise model is used. I also note that this section doesn't mention the reference frame or issues associated with inter/intra mission biases (in particular TOPEX A/B). For the later, I refer back to my overarching remarks in the top section of my review. For the former, it may be worth adding a sentence and referring to Ablain et al (2015) if appropriate.

P15 L5: The conclusion fails to mention a major issue present in the record. See overarching comments in the top section of my review.

P15 L7: Improved rather than accurate?

P15 L9: spatial scales

P15 L25: Defining GPD+ this way masks the fact it includes numerous other radiometers – perhaps worth a mention?

P15 L27: ...At the regional scale...

P15 L29: These differences are also a factor of the difference in time period, correct?

P16 L16: ...are still not reached over some specific spatial and temporal scales (e.g....

P17 L18: Spelling/font issue at end of line

P17 L21: Ref needs updating. Incorrect abbreviation as well.

P19 L33: Is this accepted yet?

P20 L4: Authors shouldn't cite in prep work. Update or remove.

P20 L33: This ref is missing some additional bibliographic info.

P22 Table 1: Make it clear that "Ensemble mean" is mean of observed GMSL. This table is also partly disingenuous as it presents the agreement in trend as evidence the product is robust. In fact, as shown by Watson et al (2015), later confirmed by Chen et al (2017), Dieng et al (2017) and Beckley et al (2017) these records are all in error. To avoid confusion, this needs to be clearly stated.

P22 Table 2: The time frame of CCI v2.0 was mentioned as 2014-2017 – the difference in date ranges here needs clarification.

P22 Table 2: What is the source of the user requirements?

P23 Figure 1 caption: ...ECV). Annual and semi-annual signals have been removed.

P23 Figure 2 Units on the colour bar. Font size here is pretty small!

P24 Figure 3 Why not combine this figure with Figure 1? (and separate that into 1 and 2?). Yellow is a poor colour choice here as it is hard to see.

P24 Figure 4 I recommend removing the lines from the legend, or at least from the

1993-2014 label as this isn't actually shown. Why does this figure only go up to 2015.0 whereas Figure 3 goes up to 2016.0?

P25 Figure 5 As per comment, is this really correct?

P27 Figure 9 This figure has a title which is inconsistent with other figures. This figure is also inconsistent with the dataset provided – it has corrections for the TOPEX issue – see comments in the top section of this review.

P29-31 Figures 11/12/13: Poor resolution in the PDF I have.

---

## Author Comment (AC1) · 21 Dec 2017

We would like to thank reviewer #1 for his/her contribution to our paper. First, an anomaly occurred in the numbering of the sections (there was two section 4). This has been corrected.

Reviewer 1, comment 1: Errors and uncertainties:

"Sources of errors are discussed in the article as well as discrepancies, however, a reader is left without an error quantification for some outputs of the data set. I would expect error bars for global sea level rise and error maps for regional trend map. Moreover, I don't see any reference to the IPCC, where sea level was well considered, e.g. will this data set contribute to the next IPCC report? "

Answer to reviewer:

Section 6 is dedicated to the mean sea level error characterization and uncertainties. We would like to stress that the first goal of this section is to make the users aware of the different contributors to the altimeter sea level errors (with associated references). An estimation of the global MSL trend uncertainty is provided (0,5 mm/yr during 1993-2015 within a confidence interval of 90% / 1.65 sigma), which can be considered as an error bar of the GMSL rise. The regional MSL trend uncertainty is discussed and the sources of errors are listed with some references. Within the SL_cci project, different studies have been carried out regarding the sources of uncertainties, errors bars, confidence envelope and inter mission biases (see for instance: http://www.esa-sealevel-cci.org/webfm_send/537 ). However, the MSL error characterization is still an activity in progress within the project with a focus on the estimate of a confidence envelope of the GMSL during the total altimetry period (error bar of the global mean sea level rise). A dedicated article will be written when results are finalised.

Text modified:

Section 5 (will be section 6) has been reworded considering this comment and the additional comments of reviewer #3. In addition, the following sentence has been added at the end of the section: "The MSL error characterization is still an on-going activity within the SL_cci project and further details regarding error bars and uncertainties will be published when results are finalised."

Answer to reviewer:

The IPCC 2013 is cited in the first sentence of the introduction, highlighting how important the sea level record has been considered. We currently do not know whether the SL_cci dataset will contribute to the next IPCC report. However, intense communication has been carried out during international conferences and with scientific publications to make the scientific community aware of this dataset. In addition, a dedicated delivery has been made to the Obs4MIPs community which aims at making observational products more accessible for the climate modelling community.

Reviewer 1, comment 2: Arctic sea level:

"Two new Arctic sea level records are mentioned and cited but nothing is showed in the article. Due to the importance of the Arctic Sea, some results have to be presented and discussed, in particular agreements/disagreements between the two products and some validation and comparison of the two products. Why the first data set does not include ERS and CryoSat?"

Answer to reviewer:

It is mentioned in section 2 (will be section 3) that two different Arctic sea level products have been developed within the SL_cci project but we have not provided more details on these products for different reasons: i) this is not the first objective of the paper, which rather focuses on the sea level ECV product (gridded sea level maps) and we wanted the article to keep a reasonable length; ii) dedicated papers have been or will be published regarding these Arctic sea level datasets: the processing of the CLS/PML product is described by Poisson et al. (2017) and the one of the DTU/TUM products is described by Passaro et al., 2017. In addition, both products have already been compared and the results can be found in Carret et al. (2016).

Text modified:

A reference to Carret et al. (2016) was missing and has been added in the text (at the end of section 3). An additional quality assessment of these Arctic sea level products has been carried out and the associated results are planned to be published soon. Thus, we have only added the reference to Carret et al. (2016).

At last, note that the CLS/PML Arctic sea level product has been generated within

some SL_cci R&D task and the time required for its development and production did not allow the ERS and CryoSat-2 data to be included.

Reviewer 1, comment 3: Access to the data:

"The data set accessible via the given DOI identifier, however, the access mechanism to data is questionable as people have to send an e-mail and get permission. I see the producers want to take track of users, but there are automatic methods to do that, while users want download easy and immediate."

Answer to reviewer:

The access to the data is for free after sending an email. Such a choice compared to a direct and immediate access has been the subject of long discussions within the CCI projects in charge of the different ECVs covered by the ESA program. Indeed, the plan was to provide a free access to the sea level data. However, ESA and the SL_cci consortium thought that it is essential to know the users in order to inform them about evolutions and above all to better answer their needs. Thus, a request for access by email has been selected.

Text modifications (typos listed by reviewer #1):

Page 4, row 23: Done.

Page 5, row 16: Done.

Page 13, row 27-29: Done.

Page 19, row 31: Passaro et al., 2017 is still under review.

Page 20, row 4: the reference Prandi et al. (2017) has been removed as it is still not published.

In addition, the overall quality of the written English has been improved following the review of co-authors. Figure 16 has been changed in landscape format and the presentation of Figure 17 has been improved. Additional evolution of the text and figures has been provided following the comments of reviewers #2 and #3.

---

## Author Comment (AC2) · 21 Dec 2017

We would like to thank reviewer #2 for his/her contribution to our paper. First, an anomaly occurred in the numbering of the sections (there was two section 4). This has been corrected.

Text modifications:

Review 2, comment Page 2, row 4: "in the Abstract, authors mentioned about the GCOS requirements. But no clear explanation is given in the Abstract nor in the Introduction. The reader is left without clear explanation about what actually are the GCOS

requirements."

Answer to reviewer:

We agree that more details on what are the GCOS requirements would be useful.

Text modified:

We have modified the end of the introduction with the following: "The sea level errors and uncertainties are discussed in Sect. 6 with respect to the GCOS requirements (GCOS, 2011). They correspond to error levels to be met by the sea level record at different spatial and temporal scales (e. g. long-term evolution, inter-annual and annual signals). These requirements have been considered as a reference within the CCI program and especially when assessing the quality of the SL_cci ECV. The paper finishes with the discussion of the perspectives of evolution of the sea level products."

Review 2, comment Page 6, row 10: "Author wrote 'In the v1.1 SL_cci ECV, a 1 mm jump was found in the GMSL around mid 2008'. Is this referred to Fig.4? If so, please indicate in the sentence."

Answer to reviewer:

This jump in the v1.1 GMSL is not explicitly shown in the paper but is partly visible in the v2.0 – v1.1 GMSL differences shown in Fig. 4.

Text modified:

This has been mentioned in the text: "In the v1.1 SL_cci ECV, a 1 mm jump was found in the GMSL around mid 2008 (partly visible in the v2.0 – v1.1 GMSL differences shown in Fig. 4)."

Review 2, comment Page 7, row 26: +3 cm2. 2 should be written in superscript.

Answer to review and text modified: Done.

Review 2, comment Page 24, Figure 3: "Does the x-axis referred to Years? If so, please

indicate. The legend should be improved. What are the solid and dashed lines in black (and dashed lines in yellow)?"

Answer to reviewer:

The x-axis is the time (years). Note that following the comment of reviewer #3, Fig. 3 is now Fig. 1.

Text modified: The figure has been updated so that it is clearer for the reader and the legend of the figure has been updated: "Figure 1: Comparison of the SL_cci v2.0 global MSL (solid orange line) with the associated linear trend (dashed orange line) and the ensemble mean (solid black line) of the global MSL derived from different groups (DUACS DT2014, CSIRO, Colorado University, GSFC and NOAA) with the associated linear trend (dashed black line) during the period 1993-2015. During this period, the trend of the SL_cci global MSL amounts to 3.3±0.5 mm/yr in a 90% confidence interval. The grey envelope shows 1.65 standard deviation of the ensemble mean (90% confidence interval). The seasonal variations have been removed and an offset has been introduced so that the mean of the 1993 data is set to zero."

Reviewer 2, comment Page 24, Figure 4: In the Figure caption and text, Author highlights a jumped about 1mm in mid-2008. Unfortunately, the jump is not obviously appeared in the Figure. I suggest the author highlights the jump area by showing it in a scar-box.

Answer to review and text modified: The figure has been adapted and the legend has been updated accordingly: "Figure 4: Global mean sea level differences between the SL_cci ECV v2.0 and v1.1. The trends are indicated for the periods 1993-2003 and 2004-2014. No trend difference is observed between ECV v1.1 and v2.0 during their common period 1993-2014 (not shown). A jump is observed in mid-2008, illustrating the anomaly of ECV v1.1 that has been corrected in ECV v2.0 (see the black box)."

In addition, the overall quality of the written English has been improved following the

review of co-authors. Figure 16 has been changed in landscape format and the presentation of Figure 17 has been improved. Additional evolution of the text and figures has been provided following the comments of reviewers #1 and #3.
* * *

---

## Author Comment (AC3) · 21 Dec 2017

We would like to thank reviewer #3 for his/her contribution to our paper which has led to a significant increase of the quality of the paper.

First, an anomaly occurred in the numbering of the sections (there was two section 4). This has been corrected.

Reviewer 3, comment #1:

"There is somewhat of a contradiction that flows through this manuscript, and remains present into the conclusions. The work claims to present an accurate GMSL record,

yet awkwardly introduces the systematic bias associated with TOPEX-A, as brought to attention by Watson et al. Nat. Clim. Chg. (2015). The work also cites Valladeau et al (2012), Dieng et al (2017) and Beckley et al (2017) around this issue, noting the omission of a citation of the relevant Chen et al., Nat. Clim. Chg. (2017) work. As the authors correctly highlight, these recent advances have led to a downward systematic revision of the rate of change in GMSL (that isn't accounted for in the CCI v2.0 dataset). Given the news piece by Tollefson (Nature News, 2017) and subsequent letters to the editor by Nerem, Cazenave and Church (Nature, 2017), it is important to avoid confusion on this issue. This manuscript describes a dataset that *does not* correct for the TOPEX related issues, yet parts of its accuracy assessment (e.g. assessment of the sea level budget, Figure 9) *does* correct for these issues (Pg8, Line 25), thus presenting a somewhat misleading perspective. The authors also present trend differences (record minus ensemble mean) as a measure of how robust the CCI record is (Table 1) – this is somewhat misleading as all of these records contain the error associated with the TOPEX record. I'm not convinced the related uncertainty section is robust without at least some discussion of issues associated with TOPEX (in particular, the A/B bias, retracking etc) I contend that these issues need to be made entirely clear is a revised manuscript. The conclusions need to include some remarks on this important issue, pointing towards the future release of a reprocessed TOPEX dataset. I would also recommend revising the title to "An improved and homogeneous... "

Answer to reviewer:

We agree that we should avoid any confusion for the reader regarding the TOPEX-A drift anomaly. This issue was initially discussed in the submitted version, but more details may be required. Thus, we propose to add the text listed below. In addition, the legend of Fig. 9 (associated with the budget closure approach) has been modified, clearly mentioning that the TOPEX-A drift correction has been applied to the GMSL specifically (and only) for this approach. In Figure 10, all records contain the error associated with the TOPEX-A drift. As the altimeter processing performed by the different

groups are not the same, this inter comparison highlights the contribution of the GMSL uncertainty associated with the data processing. This will be mentioned at the end of section 4.1 (will be 5.1). The conclusions will be modified to highlight that the SL_cci GMSL has not been corrected for the TOPEX-A drift (see the proposed text below). The title has been modified as suggested.

Text modified:

The following text has been removed from section 3.1 (will be 4.1): "Comparison with in-situ tide gauge measurements (Valladeau et al., 2012) indicates that no drift is found in this reprocessed dataset given the uncertainty of the method (Prandi et al., 2015)." "Such a drift (∼1.5 mm/yr in terms of GMSL trend) has not been corrected in the SL_cci v2.0 ECV. Applying the suggested correction would lead to a reduced GMSL trend during the total period (3.0 mm/yr) and a greater GMSL rate of rise during the second half of the altimetry era compared to the first half, highlighting that the GMSL rise is accelerating.

The following text has been added in section 3.1 (will be 4.1): The instrumental drift of the TOPEX-A altimeter is known for long (Hayne and Handcock, 1998), leading to the switch early 1999 to the redundant TOPEX-B altimeter. But until recently, it was considered that the TOPEX-A drift had minimal impact on the GMSL. Based on a comparison between TOPEX-A sea level and tide gauges data, Valladeau et al. [2012] challenged this conclusion but did not quantify this effect on the GMSL. More recently, 3 studies have attempted to quantify the effect of the TOPEX-A drift on the GMSL trend over the period January 1993-December 1998. Watson et al. (2015) compared altimetry-based sea level with vertical land motion-corrected tide gauges data and estimated a TOPEX-A drift correction to the 1993-1998 GMSL trend in the range 0.9±0.5 mm/yr to 1.5±0.5 mm/yr, with 1.5 mm/yr being the preferred value. Using a sea level budget approach, Dieng et al. (2017) also estimated the TOPEX-A drift correction to 1.5±0.5 mm/yr for 1993-1998. Another approach was followed by Beckley et al. (2017), consisting of suppressing the so-called 'internal calibration-mode range correction, included in the

TOPEX-A 'net instrument' correction, and considered as suspect. Accounting for the TOPEX-A instrumental correction for the first 6 years of the altimetry data set, these studies provided a revised GMSL time series that slightly reduces the average GMSL rise over the altimetry era (from 3.3 mm/yr to 3.0 mm/yr), but shows clear acceleration over 1993- present. Using the corrected GMSL time series, Dieng et al. (2017) and Chen et al. (2017) found improved closure of the sea level budget compared to the uncorrected data. In this paper, no TOPEX-A drift correction has been applied on the dataset available for the users as there is not yet consensus on the best approach to estimate it. However, ongoing work involving space agencies (National Aeronautics and Space Administration –NASA- and Centre National d' Etudes Spatiales –CNES) together with scientific teams should provide in the near future guidance and recommendations about this issue. As far as the SL_cci project is concerned, a corrected GMSL time series will be delivered to users in due time. "

In the discussion of Fig. 9 in section 4.1 (will be 5.1), a reference to the updated text in section 3.1 (will be 4.1) has been added.

The following sentence has been added at the end of section 4.1 (will be 5.1): Âń As the altimeter processing performed by the different groups are not the same, this inter comparison highlights the contribution of the GMSL uncertainty associated with the data processing. Âż

The following has been added in the conclusion: Âń The SL_cci GMSL has not been corrected for the TOPEX-A instrumental drift (recently highlighted by several studies). Even if several approaches have been proposed, there is not yet consensus on the best way to estimate this correction. The recommendation of the Ocean Surface Topography Science Team is to wait for the future release of a reprocessed TOPEX dataset (currently in progress by the space agencies). Âż

The legend of Figure 9 has been adapted: "Sea level budget using the SL_cci v2.0 GMSL time series (adapted from Dieng et al., 2017). The 1.5 mm/yr correction supposed to represent the TOPEX-A drift has been applied (specifically in this figure) to the GMSL time series for 1993-1998 as discussed in section 4.1 (see Dieng et al., 2017 for details)."

Reviewer 3, comment #2:

"The section on regional rate uncertainties is presently poorly evidenced. The manuscript cites work that is "in prep", and no real mention of meaningful uncertainties around regional trends are provided. This should be addressed in a revised version of the manuscript."

Answer to reviewer:

This section has been reworded considering this comment and the additional comment further in this review. The reference to the unpublished paper has been removed and a sentence has been included regarding the TOPEX-A issue. We would like to stress that the first goal of this section is to make the users aware of the different contributors to the altimeter sea level errors (with associated references). Within the SL_cci project, different studies have been carried out regarding the sources of uncertainties, errors bars, confidence envelope and inter mission biases (see for instance: http://www.esa-sealevel-cci.org/webfm_send/537 ). However, the MSL error characterization is still an activity in progress within the project with a focus on the estimate of a confidence envelope of the GMSL during the total altimetry period (which is linked to the TOPEX errors). Once finalised, the results will be published in a dedicated article.

Text modified:

Section 5 (will be section 6) has been reworded considering this comment and the additional comment further in this review.

Reviewer 3, minor issues:

P2 L4: Reference required after GCOS requirements Answer: (GCOS, 2011) has been added.

P2 L7: Global Mean Sea Level is abbreviated here as MSL but later as GMSL Answer: the abbreviation GMSL has been provided here. . . .heat added to the ocean. . . Answer: corrected

P2 L8: . . .monitoring of sea level. . . Answer: corrected

P2 L10: validation of climate models Answer: corrected

P2 L11: "supposed to correctly" isn't this implicit? I recommend removing. Answer: removed . . .altimetry mission have delivered. . . Answer: corrected

P2 L13: Suggesting adding e.g. before these references as this is not exhaustive Answer: added

P2 L17: . . .This program aims to realise the full. . . Answer: corrected

P2 L20: . . .distributed to the user community. Answer: corrected

P3 L5: I don't think these brackets are required. Answer: removed

P3 L11: The sentence beginning Thus a single. . . doesn't make sense and needs to be reworded. Answer: This has been reworded.

P3 L16: The early section defines v2.0 incorporating 2014-2017 data. This needs to be clarified – it is really 2014-2016 inclusive? If so, does it include Jason-3. This needs clarification. Answer: In the abstract, it is mentioned that the v2.0 ECV (covering 01/1993-12/2015) has been produced during the second phase of the project which took place during 2014-2017. There has been a misunderstanding between the period of the project and the period covered by the sea level record. The abstract has been adapted to clarify that the 2014-2017 period is related to the second phase of the project.

P3 L19: ...since sea level estimation from altimetry requires. . . Answer: corrected

P3 L21: Again, insert e.g. before references. Answer: done

P3 L23: ...most appropriate solution/algorithm to ensure... Answer: corrected

P3 L26: remove the word analysis Answer: removed

P3 L31: remove quotes from around pole tide Answer: removed

P4 L11: ...SL_cci products (including the SLA) (reference)... Answer: corrected

P4 L12: ...both datasets should be... Answer: corrected

P4 L31: ...for users. The... Answer: Corrected

P5 L8: ...has been carried out over different spatial and temporal scales. Answer: corrected

P5 L12: Here the CI is given as 95% but on page 14 it is 90%. Clarify. Answer: This should be 90%. This has been corrected.

P5 L12-25: See my significant remarks around how this section is articulated (and should be improved) in the top section of my review. Answer: The section has been reworded (see the answer to the first comment above).

P5 L16: ...However, over decadal time scales, the v2.0 GMSL trends are significantly different to those from v1.1... Answer: corrected

P5 L25: I notice here that Chen et al. Nature Climate Change, 2017 is not referenced, and should be. Answer: Done

P5 L25: After mentioning the 1.5 mm/yr in terms of the GMSL trend, the date period for this metric should be provided. Answer: This sentence has been removed in the revised version of this section.

P5 L28: GMSL rise is not accelerating, GMSL is accelerating. Rise is typically thought of as a rate of change. That rate is increasing, i.e. the level is accelerating. You have one too many derivatives here. Answer: This sentence has been removed in the revised version of this section. Anyway, we agree with this comment.

P6 L6: . . .ocean models. Users interested. . . Answer: Corrected

P6 L11: . . .correction, and present in the Radar. . . (This sentence is unclear, why is RADS introduced here? Clarify). Answer: This has been reworded in order to avoid any confusion.

P6 L20: . . .smoother periodic signal. . . (it is unlikely to be a pure sinusoid) Answer: The text has been adapted to avoid the 'sinusoidal' term.

P6 L21: Figure 5 – is this correct? I'm surprised by the lack of agreement in the annual terms from other labs. Please confirm this has been computed correctly using the same smoothing across all data centres. Answer: Figure 5 is correct. There is indeed a lack of agreement in the annual terms from the different groups.

P6 L26: . . .sea level difference computed against in-site data. . . Answer: corrected

P6 L32: What independent data? Clarify. Answer: The sentence has been reworded.

P7 L4: . . .both datasets. . . Answer: corrected

P7 L9: . . .Southern Ocean, the. . . Answer: corrected

P7 L16: Are these differences significant given the difference in data span? See my remarks around how regional uncertainties are addressed in the top section of my review. Answer: The periods covered by the v1.1 and v2.0 SL_cci are 1993-2014 and 1993-2015 respectively. However, the regional mean sea level trends differences between both ECVs (Fig. 7) have been computed after restricting to the common temporal coverage (1993-2014). So the data span is the same. The observed differences are mainly related to the different orbit solutions. Depending on the region considered, we think that these differences can be significant compared to the value of the sea level trend.

P7 L32: . . .variance estimated. . . Answer: corrected

P8 L14: The glacier mass term in the equation has an additional (t) in it that needs to

be removed. Answer: corrected

P8 L17: . . .refer to changes in mean glacier, Greenland and Antarctic mass balances, land water storage (LWS) and atmospheric water vapor (AtmWV). Answer: corrected

P8 L25: See my significant remarks around how this section is articulated (and should be improved) in the top section of my review. Answer: a reference to the discussion earlier in the paper has been added.

P8 L30: The use of "ensemble mean" is slightly unclear. . . in this section, is the ensemble mean referring to the sum of the terms in the budget? This needs to be clarified in captions and text. Answer: Indeed, the text was incorrect. This has been reworded. Captions are correct.

P8 L31: . . .that looking at solely the trend does not allow. . . Answer: corrected

P9 L1: This statement is somewhat disingenuous – it is in fact equal closest as the CSIRO dataset has the same difference. Answer: The text has been adapted.

P9 L3: Regarding Fig 10, around 1996-1999 it appears CU and CSIRO are quite different to the ensemble mean, but CCI is not. Interestingly, CU and CSIRO seem anticorrelated. This doesn't appear to be mentioned in the manuscript but is likely an important point worthy of a remark. Answer: This remark has been added in the text.

P9 L4: This sentence could be improved – "no anomaly" is not exactly the case (refer to the TOPEX issue, see comments at the top of my review) and "better" is a poor choice of words, consider revising. Answer: This has been reworded.

P9 L8: . . .using the NEMO. . . Answer: corrected

P9 L14: "freshwater constraining" is constraining the correct word here? Answer: modified

P9 L18: . . .which makes the estimation of GMSL (and its seasonal cycles) non trivial. Answer: corrected

P9 L26: . . .(middle panels, Fig. 11). . . Answer: corrected

P10 L25-30: This discussion does not mention regional uncertainty at all. Answer: The text has been adapted.

P11 L10: . . .gyre. Consistent with the findings presented here, it is possible to. . . . Answer: This has been adapted in agreement with the comment P11 L26.

P11: Sometimes SPG is abbreviated, sometimes it isn't, be consistent. Answer: the acronym is not used any more.

P11 L14: Is advanced the best word here? Perhaps new? Answer: corrected with 'new'

P11 L26: This repeats line 14. Revise. Answer: This has been adapted in agreement with the comment P11 L10.

P12 L34: datasets (no space) Answer: corrected

P13 L1: . . .improve much compared. . . Answer: corrected

P13 L2: one dec place to be consistent Answer: corrected

P13 L27-30: Font issue here Answer: corrected

P14 L6: . . .In Figure 18, annual. . . Answer: corrected

P14 L8: . . .represented (for model datasets, the thermos-steric component is used). Answer: corrected

P14 L10: Put see reference in brackets. Answer: corrected

P14 L13: . . .which have high negative trend values, especially. . . The especially part here is ambiguous as the sentence refers to differences between models, and then between modes and obs. Suggest reword this sentence. Answer: The sentence has been reworded.

P14 L15: Error not Errors Answer: corrected

P14 L17: estimation not estimations Answer: corrected

P14 L19: 90% or 95% as earlier indicated? Answer: 90% (95% earlier has been corrected). It has been added that this corresponds to a 1.65 standard deviation confidence interval.

P14 L25: This section is weak – there is no robust discussion around regional uncertainty. The reference is "in prep". If this is published, update, else remove and reword this section. There should be discussion around the noise model used to estimate "order of 2-3 mm/yr", as I'm sure that isn't a 90% CI value when an appropriate noise model is used. I also note that this section doesn't mention the reference frame or issues associated with inter/intra mission biases (in particular TOPEX-A/B). For the later, I refer back to my overarching remarks in the top section of my review. For the former, it may be worth adding a sentence and referring to Ablain et al (2015) if appropriate. Answer: This section has been reworded considering the major comment of the reviewer. The reference to the unpublished paper has been removed and a sentence has been included regarding the TOPEX-A issue. We would like to stress that the first goal of this section is to make the users aware of the different contributors to the altimeter sea level errors (with associated references). Improved estimation of the sea level uncertainties is an on-going activity within the project and more results are expected to be published in a near future. See also our answer to the major comment #2 of this review.

P15 L5: The conclusion fails to mention a major issue present in the record. See overarching comments in the top section of my review. Answer: Several sentences have been introduced in the conclusion regarding the TOPEX-A drift anomaly.

P15 L7: Improved rather than accurate? Answer: corrected

P15 L25: Defining GPD+ this way masks the fact it includes numerous other radiometers – perhaps worth a mention? Answer: We do not think it is worth going into more

details on this subject in the conclusion. The reader can refer to the dedicated reference provided in the text.

P15 L27: ...At the regional scale... Answer: corrected

P15 L29: These differences are also a factor of the difference in time period, correct? Answer: The time period covered by the v2.0 ECV is longer because it has been produced after the previous version of the dataset (v1.1).

P16 L16: ...are still not reached over some specific spatial and temporal scales (e.g. Answer: corrected

P17 L18: Spelling/font issue at end of line Answer: corrected

P17 L21: Ref needs updating. Incorrect abbreviation as well. Answer: updated

P19 L33: Is this accepted yet? Answer: not yet accepted

P20 L4: Authors shouldn't cite in prep work. Update or remove. Answer: removed

P20 L33: This ref is missing some additional bibliographic info. Answer: corrected

P22 Table 1: Make it clear that "Ensemble mean" is mean of observed GMSL. This table is also partly disingenuous as it presents the agreement in trend as evidence the product is robust. In fact, as shown by Watson et al (2015), later confirmed by Chen et al (2017), Dieng et al (2017) and Beckley et al (2017) these records are all in error. To avoid confusion, this needs to be clearly stated. Answer: This was not clear in the text of section 4.1 (will be 5.1) where it has been made clearer. This has been also corrected in the legend of Table 1.

P22 Table 2: The time frame of CCI v2.0 was mentioned as 2014-2017 – the difference in date ranges here needs clarification. Answer: There is a misunderstanding: the SL_cci v2.0 product covers the period 1993-2015 and it has been produced during the phase 2 of the ESA SL_cci project that took place in 2014-2017.

P22 Table 2: What is the source of the user requirements? Answer: In table 3, user requirements are from GCOS (2011). This has been added in the legend of table 3.

P23 Figure 1 caption: . . .ECV). Annual and semi-annual signals have been removed. Answer: corrected

P23 Figure 2 Units on the colour bar. Font size here is pretty small! Answer: A new version of this figure is provided.

P24 Figure 3 Why not combine this figure with Figure 1? (and separate that into 1 and 2?). Yellow is a poor colour choice here as it is hard to see. Answer: We initially wanted to clearly show SL_cci GMSL on a single figure but we agree that Figures 1 and 3 could be combined, which has been done. The map of regional MSL trends is now Figure 2 and Figure 2 (map of amplitude and phase of the annual signal) is now Figure 3. In addition, the yellow colour used of the SL_cci GMSL curve has been changed in the new Figure 1.

P24 Figure 4 I recommend removing the lines from the legend, or at least from the 1993-2014 label as this isn't actually shown. Why does this figure only go up to 2015.0 whereas Figure 3 goes up to 2016.0? Answer: This has been considered in the revised version of the figure. The timeseries ends in Dec. 2014 since this is the common period of the v1.1 (1993-2014) and v2.0 (1993-2015) dataset.

P25 Figure 5 As per comment, is this really correct? Answer: Yes, correct. This has been checked

P27 Figure 9 This figure has a title which is inconsistent with other figures. This figure is also inconsistent with the dataset provided – it has corrections for the TOPEX issue – see comments in the top section of this review. Answer: It has been clearly mentioned in the legend of Fig. 9 that a TOPEX-A drift correction has been applied.

P29-31 Figures 11/12/13: Poor resolution in the PDF I have. Answer: A 3Mo version has been provided for Fig. 11 and Fig. 12 and 13 seems fine to us.

In addition, the overall quality of the written English has been improved following the review of co-authors. Figure 16 has been changed in landscape format and the presentation of Figure 17 has been improved. Additional evolution of the text and figures has been provided following the comments of reviewers #1 and #2.